# TopoLM: Brain-like Spatio-functional Organization in a Topographic Language Model

**Neil Rathi**[*,1,2]**, Johannes Mehrer**[*,1]**, Badr AlKhamissi**[1]**,
**Taha Binhuraib**[3]**, Nicholas M. Blauch**[4]**,
**Martin Schrimpf**[1,†]
[1]EPFL, [2]Stanford University, [3]Georgia Institute of Technology, [4]Harvard University

## Abstract

Neurons in the brain are spatially organized such that neighbors on tissue often exhibit similar response profiles. In the human language system, experimental studies have observed clusters for syntactic and semantic categories, but the mechanisms underlying this functional organization remain unclear. Here, building on work from the vision literature, we develop TopoLM, a transformer language model with an explicit two-dimensional spatial representation of model units. By combining a next-token prediction objective with a spatial smoothness loss, representations in this model assemble into clusters that correspond to semantically interpretable groupings of text and closely match the functional organization in the brain's language system. TopoLM successfully predicts the emergence of a spatially organized cortical language system as well as the organization of functional clusters selective for fine-grained linguistic features empirically observed in human cortex. Our results suggest that the functional organization of the human language system is driven by a unified spatial objective, and provide a functionally and spatially aligned model of language processing in the brain.[1]

## 1 Introduction

Artificial neural network (ANN) models of language have recently been shown to accurately predict neural activity in the human language system (Schrimpf et al., 2021; Caucheteux & King, 2022; Goldstein et al., 2022). When presented with the same text input, the unit activity at internal layers of especially transformer-based models (Vaswani et al., 2017; Radford et al., 2019) is strikingly similar to the internal activity measured experimentally in human cortex. The most powerful models predict even close to 100% of the explainable variance of neural responses to sentences in some brain datasets (Schrimpf et al., 2021). However, while there is a strong alignment to the brain's *functional responses*, a crucial element of cortex is entirely lacking from today's language models: the *spatial arrangement* of neurons on the cortical surface.

In recent models of the visual system, the introduction of *topography* has led to ANNs that begin to match brain activity functionally as well as spatially (Lee et al., 2020; Margalit et al., 2024; Keller et al., 2021; Blauch et al., 2022; Lu et al., 2023). These models provide a principle for understanding the development of spatial organization in the brain, in the form of minimizing wiring cost, such that neurons with similar response profiles tend to cluster together. These clusters resemble the spatio-functional organization in the early cortex with orientation preferences such as pinwheels (Hubel & Wiesel, 1962; 1968; Maunsell & Newsome, 1987; Felleman & Van Essen, 1991), and in higher-level visual regions with category-selective regions such as face patches (Kanwisher et al., 1997; Haxby et al., 2001; Tsao et al., 2003; 2006; 2008; Freiwald et al., 2009).

The topography of the human language system on the other hand lacks a comprehensive computational explanation. Neuroscience experiments suggest both a macro-organization at the level of a distributed cortical network that selectively responds to linguistic processing (Fedorenko et al., 2010; 2011; 2024; Blank et al., 2014), as well as a micro-organization into clusters that correspond

---

[*]Equal contribution by NR and JM. [†]Correspondence: `martin.schrimpf@epfl.ch`

[1]Code available at `https://github.com/epflneuroailab/topolm`.

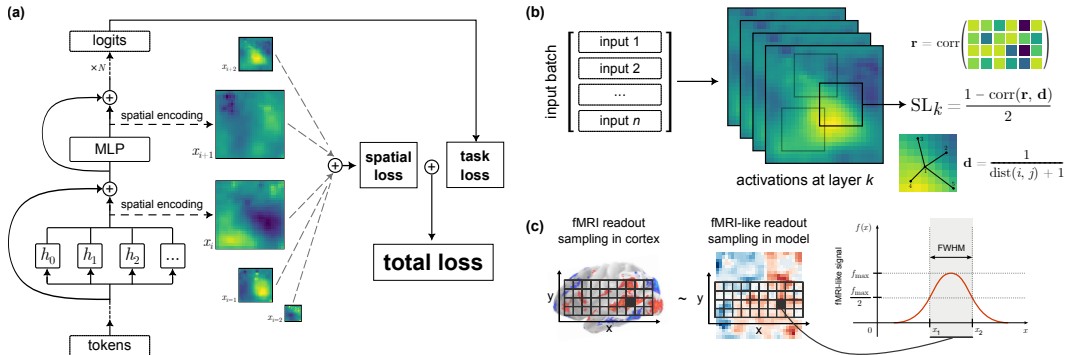

Figure 1: **Building a topographic language model with brain-like spatio-functional organization.** **(a)** TopoLM modifies the Transformer architecture with a two-dimensional spatial encoding at the output of each attention and MLP layer. This representation enables the use of a spatial correlation loss that encourages smooth response profiles in adjacent units. This spatial loss is jointly optimized with cross-entropy task loss during training. **(b)** At each forward pass, we randomly select five neighborhoods in each layer (of which we here only show 3 for clearity) and compute the pairwise correlation of unit activations within each neighborhood. The spatial loss is computed by comparing these correlations to the inverse distances between associated unit pairs, with the final loss averaged across units pairs and neighborhoods. Computing the loss on cortical neighborhoods is an efficient approximation of the spatial loss. **(c)** We use a FWHM filter to simulate the fMRI sampling process such that a simulated voxel's response ('fMRI-like signal') is composed of a combination of responses from neighboring units (Kriegeskorte et al., 2010). We simulate the response as a Gaussian random variable, with FWHM 2.0 mm, assuming unit distances of 1.0 mm.

to syntactic and semantic categories such as verbs, nouns, and concrete words (Shapiro et al., 2006; Moseley & Pulvermüller, 2014; Hauptman et al., 2024). What are the mechanisms underlying this spatio-functional organization of the language system in the brain?

Here, we develop **TopoLM**, a neural network model for brain-like topographic language processing. TopoLM is based on the transformer architecture but incorporates an explicit spatial arrangement of units. We train the model via a combined task and spatial loss, which optimizes the model to perform autoregressive language modeling while encouraging local correlation, similarly to a recent approach used in vision (Lee et al., 2020; Margalit et al., 2024). The spatio-functional organization that emerges in this model is semantically interpretable and aligned with clusters that have been observed experimentally in brain recordings. Comparing TopoLM with a non-topographic baseline model (i.e. one trained without spatial loss) on a series of benchmarks, we show that while TopoLM achieves slightly lower scores on some behavioral tasks (BLiMP), its performance on other downstream tasks (GLUE) and on brain alignment benchmarks (using the Brain-Score platform) is on par with the non-topographic control.

Importantly, this spatio-functional organization arises purely as a result of the combined task and spatial loss, as the model is trained solely on naturalistic text *without* fitting to brain data. This work thus extends the principle of cortical response smoothness proposed in vision (Margalit et al., 2024) into the language system, providing a unified explanation for understanding the functional organization of cortex.

## 2 RELATED WORK

**Topographic Vision Models.** In contrast to the core human language system, the primate visual cortex shows a clear hierarchy of interconnected regions starting at the primary visual cortex (V1), passing V2 and V4, and reaching inferior temporal cortex (IT) which is thought to underlie representations of complex visual objects such as faces and scenes. Within V1, orientation-selective cortical patches ('hypercolumns') are spatially arranged in circular 'pinwheels,' where the preferred orientation of neurons rotates smoothly around a central point, covering all possible orientations (0 to 180 degrees). This structure is observed across species, including humans and non-human

primates (Kaschube et al., 2010). On a more global level, early visual areas (V1, V2) show strong retinotopic organization where nearby stimuli in the visual field activate nearby locations in early visual regions (Engel et al., 1994; 1997; Tootell et al., 1998). While the strength of retinotopic organization strongly decreases, but remains detectable going to higher-level regions of the visual cortex (Larsson & Heeger, 2006; Schwarzlose et al., 2008; Kravitz et al., 2010; Groen et al., 2022), the final stage of the ventral visual pathway, IT, shows clear categorical clustering into e.g. regions selective for faces or scenes (Kanwisher et al., 1997; Haxby et al., 2001).

This spatial organization of the primate visual cortex has prompted work on topographic ANNs for vision. First approaches focused on the organization of inferotemporal cortex, restricting topographic organization to later model layers (TDANN, Lee et al. 2020; ITN, Blauch et al. 2022; DNN-SOM, Zhang et al. 2021, Doshi & Konkle 2023). Recent models are designed such that all layers are topographic and thus mimic topographic features across the visual cortex—for example, smoothly varying orientation preference maps forming pinwheels in model V1 and category-selective regions in model IT (All-TNNs, Lu et al. 2023; new version of TDANN, Margalit et al. 2024).

Our topographic language model belongs to the **Topographic Deep Artificial Neural Network** (TDANN) family of models (Lee et al., 2020; Margalit et al., 2024). Herein, a central claim is that inducing a preference towards *smoothness* of cortical responses in the model provides a **unifying principle** for the development of topography in the brain. This smoothness optimization is applied to all layers in the model and replicates functional organization in early (e.g. V1) *and* later (e.g. IT) regions of the visual cortex. The TDANN's spatial smoothness, as implemented with an additional loss term, is an indirect but efficient approach to minimizing local wiring-length, and can additionally help to minimize long-range connectivity which, in neuroscience terms, corresponds to brain size and power consumption (Margalit et al., 2024).

**Topographic Language Models.** Comparatively little work has explored the idea of inducing topography in language models. In particular, the only topographic language model we are aware of is BinHuraib et al. (2024)'s **Topoformer**, which induces spatial organization onto a single-headed attention Transformer architecture using local connectivity constraints. This model arranges keys and queries on 2D grids, combined with a locally connected layer in the attention mechanism as opposed to full connectivity.

Our approach primarily differs from Topoformer in that we use a spatial smoothness *loss* term to drive the emergence of local correlations, similarly to Lee et al. (2020) and Margalit et al. (2024)'s TDANN vision models. In this sense, our model extends Margalit et al. (2024)'s unifying principle of functional organization from the visual cortex into the language system. TopoLM is thus able to benefit from full connectivity, rather than requiring local connectivity to develop clustering. Because we apply this loss to the output of entire attention mechanism at each layer (as well as to the MLP), TopoLM can also benefit from multi-head attention, which empirically improves fits to neural data (AlKhamissi et al., 2024); this was not explored in (BinHuraib et al., 2024). Finally, our model also uses an autoregressive task loss, rather than a masked autoencoder objective (as used in BinHuraib et al. (2024)), which has been shown to have higher performance on neural alignment benchmarks (e.g. Schrimpf et al., 2021).

## 3 MODEL DESIGN AND VISUALIZATION

Instead of the convolutional neural network architecture used in topographic vision models (Margalit et al., 2024), we use the Transformer architecture (Vaswani et al., 2017) which is dominant in language modeling. We augment the objective function with a spatial correlation loss, in addition to the cross-entropy task loss. This loss function measures spatial smoothness, which serves as an efficiently computable proxy for neural wiring length: neurons located close to one another should have similar response profiles—i.e. their activations should be correlated (Lee et al., 2020).

To introduce a notion of 'space' in the model, we bijectively map the units of each attention layer and MLP to a square grid. We randomly permute these positions for each layer such that each layer has a unique spatial encoding.[2] On each forward pass, we first compute the pairwise Pearson's

---

[2]Our goal is to abstract away from feed-forward propagation as much as possible, as the hierarchical organization of the brain is quite different from that of a language model. This random permutation prevents the

correlation vector $\mathbf{r}_k$ between unit activations on the input batch for each layer $k$ (see Figure 1B). If a layer has $N$ units, $\mathbf{r}$ is of dimension $\text{nCr}(N, 2)$. Then, the spatial loss for layer $k$ is given by

$$\text{SL}_k = \frac{1}{2}\left(1 - \text{corr}(\mathbf{r}_k, \mathbf{d}_k)\right), \tag{1}$$

where $\mathbf{d}_k$ is a vector of pairwise inverse distances between units, based on their spatial encoding, and $\text{corr}$ is Pearson's $r$. This means that nearby units (i.e. high inverse distance) should have highly correlated activations on the same inputs, and that distant units should be less correlated; this gives us a notion of spatial smoothness. We scale by a factor of $0.5$ to ensure that $\text{SL} \in [0, 1]$.

We compute this spatial loss for every attention and MLP layer in a Transformer, prior to normalization and addition into the residual stream. Rather than computing the spatial loss for the entire layer, as in Margalit et al. (2024) we approximate the loss using small neighborhoods, ensuring that the model optimizes for local, rather than global 'long-distance' constraints.[3] For each batch of inputs, the model is then optimized subject to the loss criterion

$$\ell = \text{TL} + \sum_{k \in \text{layers}} \alpha_k \text{SL}_k, \tag{2}$$

where $\text{TL}$ is the task loss and $\alpha_k$ is the relative weight of the spatial loss associated with layer $k$. This combined loss metric encourages the model to learn representations that are both spatially organized and useful (and, in the case of self-supervised cross-entropy task loss, task-general).

**Model Specification and Training.** In the below experiments, we utilize an adapted GPT-2-small style architecture (Radford et al., 2019). We use hidden dimension 784 such that we can evenly embed units in a $28 \times 28$ grid. The model has 12 Transformer blocks, each with 16 attention heads and a GELU activation function. We train our models on a randomly sampled 10B-token subset of the FineWeb-Edu dataset. The task loss is cross-entropy on next-word prediction. We use batch size 48 and block size 1024. For spatial loss, we set $\alpha_k = 2.5$ across all layers[4] and operationalize the inverse distance vector $\mathbf{d}$ with the $\ell^\infty$ norm. For each batch, we average the spatial loss across 5 randomly selected neighborhoods, each of $\ell^\infty$ radius 5. This allows us to compute loss more efficiently without significant performance drops.

We train both a topographic model and a non-topographic baseline, where $\alpha_k = 0$ and all other hyperparameters remain the same.[5] We trained both models with early stopping after three consecutive increases on validation loss. At the end of training, the topographic model achieved a validation task loss of 3.075 and spatial loss of 0.108 (summed across layers), while the non-topographic model achieved validation loss 2.966. Models trained for 5 days on 4xNVIDIA 80GB A100s.

In all below analyses, we compare TopoLM to BinHuraib et al. (2024)'s pre-trained Topoformer-BERT, a BERT-style model (Devlin et al., 2019) with local connectivity trained on the BookCorpus dataset (Zhu et al., 2015).[6] Note critically that Topoformer-BERT is a *baseline*, but not a control—it is trained on a much smaller corpus, has only one attention head per layer, and is bidirectional.

---

model from exploiting the feed-forward nature of the Transformer. Without it, the model minimizes spatial loss by propagating the same spatial pattern through the network; see Figure 12.

[3]This could be controlled for using many alternative methods, e.g. a Gaussian smoothing kernel. However, we choose to use neighborhood-level approximations for simplicity.

[4]We chose this value of $\alpha$ after extensive hyperparameter search. In particular, lower values of $\alpha$ do not adequately encourage the development of topography, while greater values impede task performance and the development of meaningful representations.

[5]After hyperparameter tuning, we optimize using AdamW with $\beta_1 = 0.9, \beta_2 = 0.95$ and learning rate $6 \times 10^{-4}$, scheduled with warmup and cosine decay. We use weight decay 0.1, gradient clipping at 1.0, and do not use dropout. Neighborhood size and number of neighborhoods were also determined via hyperparameter search; note, however, that we empirically observed little effect of neighborhood size and number of neighborhoods on task performance or on our topographic metrics.

[6]Topoformer-BERT has 16 layers, each with a single attention head, with topographic constraints applied both at the output of each attention layer and after the key matrix product; we therefore compare TopoLM to outputs at these levels. We refer to BinHuraib et al. (2024) for more details.

**(a)** TopoLM: Individual Responses Across Language Selective Clusters

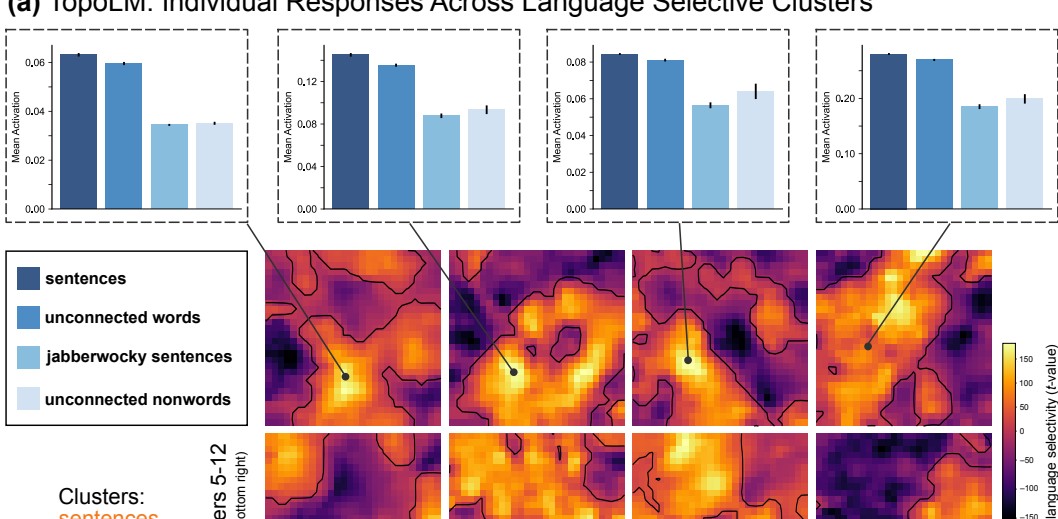

**(b)** Fedorenko et al.: Individual Responses Across Anatomical Regions

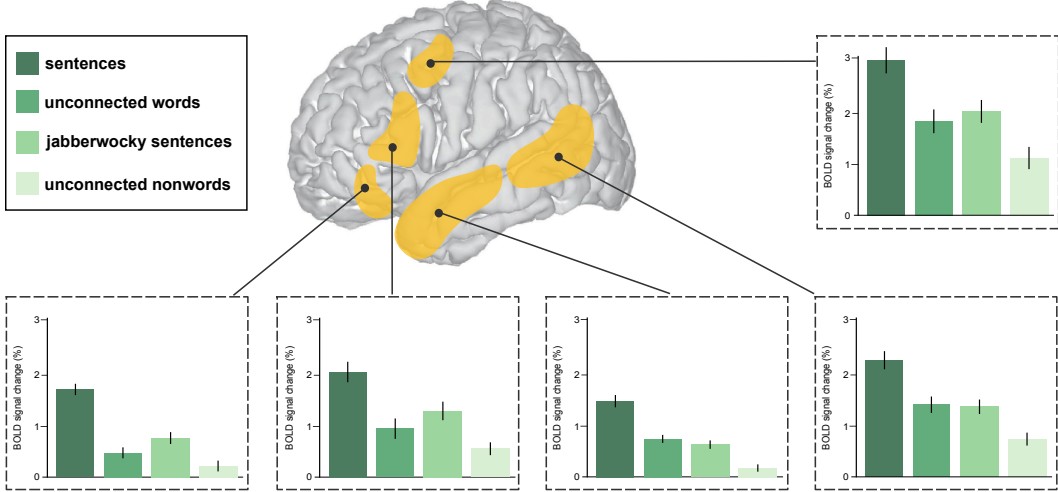

**(c)** Whole-network Responses in TopoLM, Non-topographic Baseline, and Neural Data

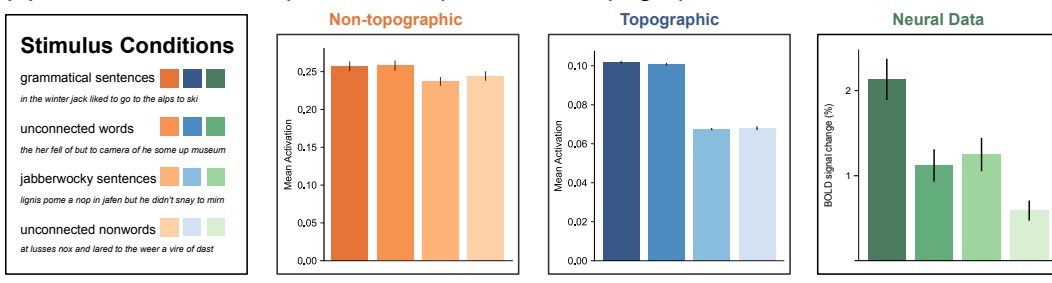

Figure 2: **Brain-like response profiles across the core language system. (a)** Applying a functional localizer (Fedorenko et al., 2010) we isolate the core language system of TopoLM, and find clear brain-like spatial organization (for brevity, we only show Transformer blocks 5-12 here). Response profile across individual language-selective clusters (shown in yellow) in TopoLM are similar to one another, consistent with **(b)** the language system in human cortex (Fedorenko et al., 2024). **(c)** Across the entire core language system, TopoLM (blue) *mostly* matches the neural data (green), but not exactly; however, the non-topographic baseline model (orange) fails to capture neural patterns as well.

**Readout Sampling.** Due to the coarse spatial sampling in fMRI neuroimaging work, voxels contain the aggregated response of a large population of neurons (Kriegeskorte et al., 2010, Figure 1C). In all following analyses, we thus apply a simulated version of fMRI readout sampling to model activations, consisting of smoothing with a Gaussian kernel, to imitate the locally aggregated responses of fMRI voxels. Importantly, we do so before computing selectivity based on these activations, and thus do not apply readout sampling to the functional selectivity maps directly. We set unit distance 1.0 mm and FWHM 2.0 mm.

# 4 SPATIO-FUNCTIONAL ORGANIZATION OF THE CORE LANGUAGE SYSTEM

Language processing in the brain engages a set of left-lateralized frontal and temporal brain regions. These areas are typically referred to as the 'core language system' (Fedorenko et al., 2010) and respond selectively to linguistic input in contrast to non-linguistic stimuli (see Fedorenko et al., 2024, for an overview). Due to anatomical differences between individuals, the language system is defined via a **functional localizer** that contrasts syntactically and semantically valid sentences against a perceptually matched control, such as strings of nonwords (Fedorenko et al., 2011).

Within individuals, the core language system shows clear spatio-functional organization, wherein language selective neurons cluster together across multiple cortical lobes. Anatomically distinct **subregions** of this system exhibit highly consistent response profiles to stimuli, suggesting that the system operates as a network (e.g. Fedorenko et al., 2011; Tuckute et al., 2024) (Figure 2B).

Prior work on neural alignment in language models typically compares neural responses across the *entire* core language system of the brain to model activations. The topographic organization of our model enables us to test for the emergence of a brain-like spatially organized core language system *in silico*. A successful spatio-functional alignment between brain and model would mean that (1) distinct language-selective clusters emerge in the model, (2) these clusters all have consistent response profiles similar to consistent response profiles across sub-regions of the 'core language system' in humans [7], and (3) the response profiles match the activity profiles in the brain (*sentences > {unconnected words, jabberwocky} > nonwords*; AlKhamissi et al., 2024).

**Methods.** To isolate the core language system in TopoLM, we use the same localization stimuli as Fedorenko et al. (2010), which consists of a set of 160 sentences and 160 strings of non-words, all 12 words each. After passing these through the model, at each attention and MLP layer we run a $t$-test across the activations of all layer units. We then define the core language system as all units that are significantly language-selective ($p < 0.05$ after correction for multiple comparison across all layers using the false-discovery-rate (FDR) (Benjamini & Hochberg, 1995)).

We then define language-selective clusters using an evolutionary clustering algorithm applied to each contrast map. In each layer, we begin with the most selective unit by $t$-value, and then repeatedly add the most selective neighboring unit to the cluster, until we hit a pre-determined $p$-value threshold associated with the unit's selectivity (here, $p(\text{FDR}) < 0.05$). We repeat this process, searching for new clusters until all units in the layer are exhausted. We discard clusters with fewer than 10 units.

Within each cluster, we measure responses to the same stimuli used in neuroscience experiments (Figure 2B, Fedorenko et al., 2011): *sentences* (indexing syntactic and lexical information), *unconnected (scrambled) words* (lexical but not syntactic), *Jabberwocky sentences* which are well-formed sentences where content words are replaced by phonotactically plausible non-words (syntactic but not lexical), and *unconnected (scrambled) non-words* (neither syntactic nor lexical). Note that these stimuli are distinct from those used for localization. We measure the model 'response' as the mean absolute activation across all units in a cluster.

**Results.** TopoLM exhibits clear brain-like spatial organization of the language network, such that (1) multiple language-selective clusters emerge across the topographic tissue (Figure 2A and Figure 8), (2) across most clusters, the response profiles are consistent with one another (Figure 9), and (3) response profiles *mostly* match the ones in the brain. The response profiles are not a perfect match to the brain data—while sentences have higher activations than Jabberwocky and nonword stimuli,

---

[7]"The language areas all show a similar response profile (despite slight apparent differences, no region by condition interactions come out as reliable, even in well-powered studies." Fedorenko et al. (2024),

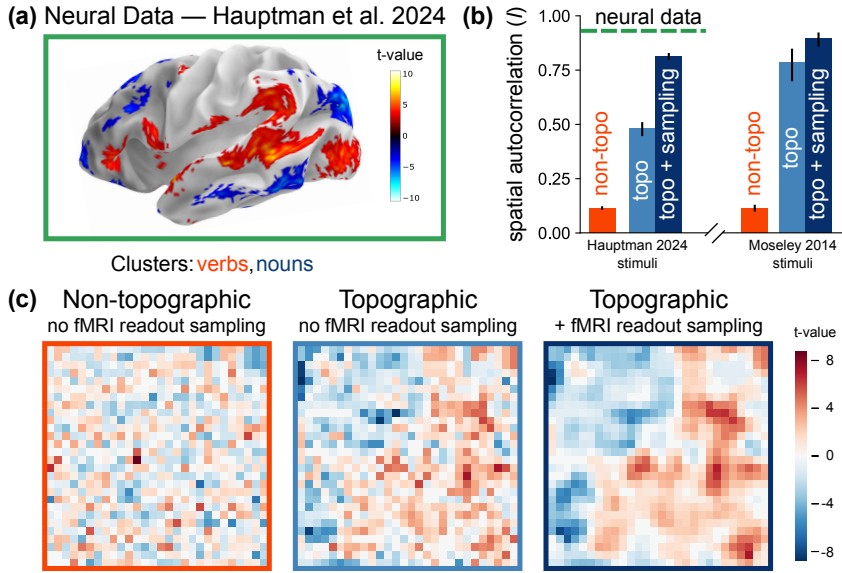

Figure 3: **Brain-like verb- and noun-selective clusters in TopoLM. (a)** fMRI data from Hauptman et al. (2024) points to verb- (red) and noun-selective (blue) regions in the human cortex with strong clustering (Moran's $I = 0.96$). **(b)** Quantification of clustering. Relative to high clustering in the brain (green dashed line), the non-topographic baseline shows limited clustering (orange). The topographic model shows moderate clustering at the unit level (light blue) and strong clustering when simulating fMRI sampling (dark blue). On stimuli from Moseley & Pulvermüller (2014) (fMRI data not available) we find qualitatively similar results. **(c)** Exemplary model maps (last MLP layer) showing the verb-/noun contrast (red-blue) in response to stimuli from Hauptman et al. (2024). The non-topographic baseline shows no clustering while the topographic model develops verb- and noun-selective clusters.

they do not have higher activation than unconnected words as in brain data. However, looking across the entire language selective network, the response profile of the non-topographic baseline model similarly fails to capture the neural response profile (Figure 2C), suggesting a general shortcoming of the base transformer model, rather than a weakness of topography. We similarly find evidence for language-selective clustering in Topoformer-BERT (see Figure 13).

## 5    SPATIO-FUNCTIONAL ORGANIZATION OF SEMANTIC CLUSTERS

Beyond selectivity for language in general, experimental evidence supports the existence of cortical noun- and verb-selective clusters across human subjects during processing of verbal and nominal stimuli in auditory (Elli et al., 2019; Hauptman et al., 2024), visual (Moseley & Pulvermüller, 2014), and speech production (Shapiro et al., 2006) tasks. Here, we compare spatial activation patterns predicted by TopoLM to two groups of fMRI studies: Elli et al. (2019) / Hauptman et al. (2024), who use the same set of stimuli and the same experimental setup (Figure 3 and Appendix Figures 7 and 10), and Moseley & Pulvermüller (2014) (Figure 4). We evaluate TopoLM predictions quantitatively using Hauptman et al. (2024)'s fMRI data[8] and perform qualitative evaluations where no neural data was available (Elli et al., 2019; Moseley & Pulvermüller, 2014).

### 5.1    CLUSTERS SELECTIVE FOR VERBS AND NOUNS

**Neuroimaging Study.**    Using fMRI data from Hauptman et al. (2024) (Appendix B), we find verb- and noun-selective clusters in the left hemisphere (Figure 3A and Appendix Figure 7), thus replicating their results. To quantify the 'degree' of clustering in these maps, we use Moran's $I$ with

---

[8]Available on OPENICPSR at `https://doi.org/10.3886/E198163V3`.

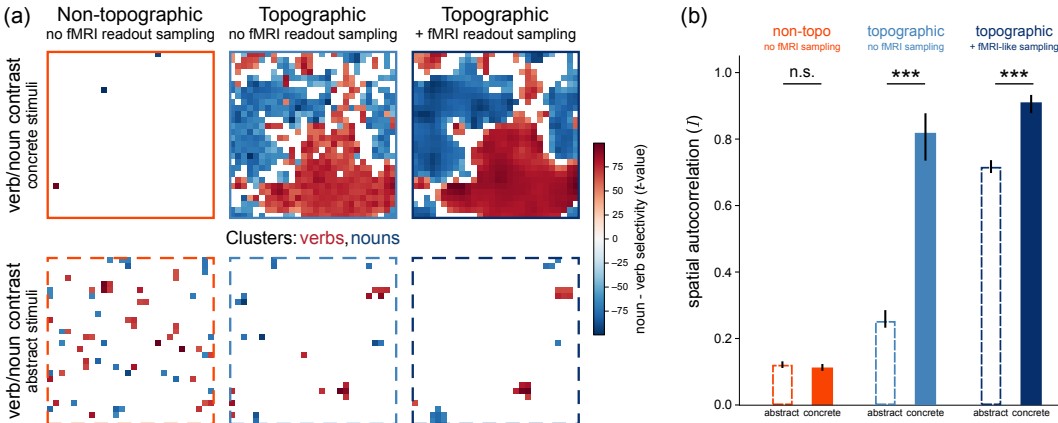

Figure 4: **Verb- and noun-selectivity in response to concrete and abstract stimuli.** **(a)** Using stimuli from Moseley & Pulvermüller (2014), we find verb-/noun-selective clusters (verb: red / noun: blue) emerging in TopoLM for concrete (solid lines), but not for abstract words (dashed lines), thus replicating their results. **(b)** We obtain strong verb-/noun-clustering when concrete words are used to compute the verb-/noun-contrast (light blue, solid lines), $I = 0.80$, but substantially lower clustering for abstract words ($I = 0.23$, light blue, dashed lines, $t$-test: $p < 0.001$). However, we do not find evidence for such a difference in verb-/noun-clustering when using the non-topographic control ($I = 0.11$ vs. $0.12$, $t$-test: $p > 0.05$, orange). Results do not change qualitatively when fMRI readout sampling is performed before computing contrasts and clustering (dark blue). In all the presented cases, spatial autocorrelation is computed on un-thresholded maps (for all layers, see Figure 11). Following neuroimaging conventions on defining category-selective cortical clusters, we show model maps thresholded at $p(\text{FDR}) < 0.05$.

Queen contiguity (i.e. $\ell^\infty$ radius, following the distance metric used to train TopoLM), a common measure of spatial autocorrelation, ranging from $-1$ to $1$ (see Appendix A for details). The group level effects indicate strong clustering ($I = 0.96$, $p < 0.001$).

**Clustering in TopoLM.** To investigate whether similar verb- and noun-selective clusters emerge in TopoLM, we extracted the model activations in response to the same stimuli as Elli et al. (2019) and Hauptman et al. (2024). We find that verb and noun-selective clustering emerges across layers in TopoLM after contrasting activations to verb and noun stimuli (verb: red / noun: blue, Figure 3A, Figure 10). Similar to our analysis of the neuroimaging data, we quantified this observation using Moran's $I$. The non-topographic baseline model yields a low degree of clustering ($I = 0.11$). Contrast maps in TopoLM show a strong degree of clustering ($I = 0.48$) which is further increased when applying fMRI-like readout sampling ($I = 0.81$, Figure 3B). Applying the same sampling to the non-topographic model also increases clustering, but it remains substantially less brain-like than the topographic model ($I = 0.60$, Appendix Figure 10).

**Clustering in Topoformer-BERT.** Applying the same procedure to Topoformer-BERT, we find no evidence for noun-verb selective clustering, with few units coming out significant in the noun-verb contrast (10.61% of units; see Figure 13). However, we do find that, before thresholding for significance ($p < 0.05$), the model does exhibit a high degree of clustering competitive with TopoLM (Moran's $I = 0.66$ before sampling, $0.85$ after sampling). In other words, though local connectivity constraint induces clustering in the model, these clusters do not match the spatio-functional organization of the brain. This impression is confirmed by additional anaylsis using a variant of Moran's $I$ that only considers units with significant t-values (Figure 14).

## 5.2 Clusters selective for concrete, but not abstract verb-noun contrasts

**Neuroimaging Study.** Moseley & Pulvermüller (2014) focus on how cortical noun-verb selectivity relates to semantics, in particular focusing on *concreteness*. Examining specific anatomically defined brain regions in fMRI, this study finds evidence for selectivity between concrete verbs and

concrete nouns; yet, critically, there is no evidence for responses to abstract words from the same categories. Here, we investigate whether TopoLM replicates these findings—both the existence of spatially organized noun-verb selectivity in response to concrete words and the *non*existence of this selectivity in response to abstract words (Figure 4).

**Clustering in TopoLM.** We presented the original experimental stimuli to TopoLM and computed contrasts between abstract and concrete verbs and nouns (see Appendix C for stimulus details). Rather than comparing response profiles in anatomically defined subregions as in Moseley & Pulvermüller (2014) (since the model lacks defined 'anatomical' regions), we explore whether lexical class-selective clustering emerges between concrete and abstract words. Consistent with brain data, we find clustering of verb- and noun-selective model units in concrete stimuli, but find only very weak or no clustering for the same contrast in abstract stimuli. We quantified this impression using again Moran's $I$: concrete words yield substantially higher verb-/noun-clustering than abstract words ($I = 0.8$ vs. $I = 0.23$ in unthresholded maps for TopoLM, Figure 4, light blue). We obtain qualitatively similar results when simulating the fMRI sampling process before computing contrasts (Figure 4, dark blue). Additionally, we observed overall low degrees of clustering in non-topographic baseline models and no evidence for a difference in the degree of clustering when concrete vs. abstract words were used for computing the verb-/noun-contrast (Figure 4, orange).

**Clustering in Topoformer-BERT.** We again apply the same procedure to Topoformer-BERT and find no evidence for noun-verb selective clustering, with no units coming out significant in the noun-verb contrast (see Figure 13). Again, before thresholding, we find high clustering (concrete-concrete: Moran's $I = 0.60$ before sampling, $I = 0.84$ after; abstract-abstract: $I = 0.61$ before sampling, $I = 0.84$ after); yet since none of these units come out significant, we fail to find evidence for brain-like spatio-functional organization in Topoformer-BERT (for details, see Figure 14).

## 6    Downstream Performance and Brain Alignment

Some topographic vision models sacrifice task performance for the sake of spatial organization, leading to diminished utility as a model of brain function (Lee et al., 2020). As a control, we evaluate TopoLM's performance on linguistic knowledge (BLiMP), downstream task (GLUE), and brain alignment benchmarks (Brain-Score), compared to the non-topographic baseline model.

**Benchmark Description and Methods.** **BLiMP** (Warstadt et al., 2020) is a multi-task linguistic benchmark consisting of minimal pairs (acceptable / unacceptable) that probe knowledge of linguistic phenomena; Models are evaluated without fine-tuning. **Brain-Score Language** (Schrimpf et al., 2021; 2020) is a set of benchmarks that measure how well models align with the human language system. We train a ridge regression model to predict brain activity from model representations, using the same stimuli as in the neuroimaging studies. Representations are taken from each transformer block with spatial loss (attention and MLP). For both models, we select the output layer with the best alignment across datasets. The final score is the Pearson correlation between actual and predicted brain activity, averaged over 10 cross-validation folds and over four brain-recording datasets (Blank et al., 2014; Fedorenko et al., 2016; Pereira et al., 2018; Tuckute et al., 2024, Appendix D). **GLUE** (Wang et al., 2018) is a multi-task benchmark for downstream performance on tasks like entailment and sentiment analysis. We fine-tune models for each task. During fine-tuning, we use the same weighted combination of task loss (cross-entropy or MCC) and spatial correlation loss with $\alpha = 2.5$. We set batch size 64, dropout 0.1, and learning rate $2 \times 10^{-5}$; all other hyperparameters are the same as during pretraining. We use early stopping after 3 successive increases in validation loss, keeping the checkpoint before this increase.

**Results.** We find that introducing spatial correlation loss does not strongly affect task performance or brain alignment (Table 1). On BLiMP, we find slight decreases in performance between TopoLM and the non-topographic baseline across all subtasks, with a 5 point average decrease overall. On Brain-Score, we find a 2 point average decrease overall, with TopoLM *out*performing the non-topographic baseline on some subtasks (see Appendix D, Figure 6). On GLUE, across most subtasks we see an *increase* in performance from TopoLM compared to the baseline model, with a 3 point average increase overall. We hypothesize this increase might be due the spatial loss term serving as additional regularization against overfitting.

| Model | BLiMP | GLUE | Brain-Score |
|---|---|---|---|
| TopoLM | 0.71 | **0.68** | 0.78 |
| Non-topo | **0.76** | 0.65 | **0.80** |

Table 1: Results from evaluation of TopoLM and a non-topographic baseline ('non-topo') on BLiMP, GLUE, and Brain-Score. The baseline model outperforms TopoLM on BLiMP (5 pts) and Brain-Score (2 pts), but TopoLM outperforms the non-topographic control on GLUE (3 pts).

## 7 DISCUSSION

Here, we presented a new topographic Transformer language model based on a spatial smoothness constraint and showed that it predicts key patterns of spatio-functional organization from the neuroimaging literature. We demonstrated that TopoLM has a spatially organized core language system exhibiting brain-like uniformity of response profiles across clusters. TopoLM also shows brain-like noun-verb selective clusters, specific to concrete over abstract words. This spatial correspondence to the human brain comes at virtually no cost to performance or functional brain alignment.

**Spatial smoothness principle.** TopoLM is in the TDANN family of models. Margalit et al. (2024) originally posited the loss term underlying TDANN models as a unifying account of the development of spatio-functional organization in the visual cortex. TopoLM successfully extends the corresponding spatial loss to the domain of language neuroscience, and thus provides evidence that this principle of spatial smoothness indeed generalizes across cortex.

**Comparison with Topoformer-BERT.** We also observe evidence for spatial organization with noun-verb contrasts in TopoLM, but not BinHuraib et al. (2024)'s Topoformer-BERT. However, Fedorenko et al. (2010)'s stimuli *do* reveal a spatially organized language network in Topoformer-BERT. This asymmetry is likely in part because the model induces topography only in the attention layers, which primarily track relationships between tokens: while Fedorenko et al. (2010)'s stimuli consists of entire sentences, the noun-verb contrast stimuli are one or two word items.

However, it's important to note that—while this comparison is a useful baseline—our primary goal is to ask whether the TDANN spatial smoothness principle generalizes to language, not to build a topographic language model that "improves" on certain benchmarks. Topoformer asks a similar question using a different guiding principle (local connectivity), but these models do not necessarily "compete" with one another: indeed, TopoLM's spatial smoothness and Topoformer's explicit local connectivity are both closely related to the principle of wiring length minimization.

**Limitations.** TopoLM is a feed-forward model of linguistic processing with multiple layers. Since the topographic maps are specific to each layer, there is as such no coherent tissue across the entire system as in the the brain. This also limits eventual modeling of 'BMI' tissue perturbations, such as micro-stimulation, since the spread of current ends at one topographic map (out of several).

**Model-guided experiments.** Since TopoLM is stimulus-computable, it can be used to discover new candidates for spatial clustering in linguistic processing. For instance, the automatic identification of clusters emerging from model activations in response to extensive textual input could inform the selection of optimal stimuli for further neuroimaging experiments. Given that the model successfully predicted spatio-functional organization in three existing brain datasets, it is likely that additional predictions derived from this approach would be informative.

Taken together, our results suggest that the spatial smoothness principle leads to topographic organization consistent with the spatio-functional organization of linguistic processing in the brain.

## 8 ACKNOWLEDGMENTS

We thank Miriam Hauptman and Marina Bedny for providing fMRI data and for useful discussion. We also thank Greta Tuckute, members of the EPFL NeuroAI lab, and members of the Stanford CLiMB Lab for helpful feedback. Neil Rathi was supported by a Summer@EPFL Fellowship.

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

## A  MORAN'S $I$ AND SPATIAL AUTOCORRELATION

We use Moran's $I$ to estimate the degree of clustering in TopoLM layer maps and in surface-based fMRI data. Moran's $I$ is given by

$$I = \frac{N}{W} \cdot \frac{\sum_{i=1}^{N} \sum_{j=1}^{N} w_{ij}(x_i - \bar{x})(x_j - \bar{x})}{\sum_{i=1}^{N} (x_i - \bar{x})^2},$$

where $N$ is the number of units (e.g. fMRI vertices or model units), $x_i$ is the value at position $i$, $\bar{x}$ is the mean of all $x_i$, and $w_{ij}$ is a weight indicating the spatial relationship between units at positions $i$ and $j$ such that $w_{ij} = 1$ if two points are 'neighbors' and $w_{ij} = 0$ otherwise (see below on how this is defined).

Moran's $I$ ranges from $-1$ to $1$. Positive values indicate spatial clustering, where similar map values (e.g., high-high or low-low) tend to group together; negative values indicate spatial dispersion, where dissimilar map values (e.g., high-low) are neighbors; and a value close to $0$ indicates no autocorrelation, i.e. a random distribution. In our data, we do not see significantly negative values of $I$, as spatial dispersion is a systematic non-random pattern not encouraged by topography.

**Application to TopoLM layer maps.**  To quantify the degree of clustering in TopoLM maps (figures 3, 4), we apply Moran's $I$ using 'Queen' contiguity (i.e. an $\ell^\infty$ radius). Queen contiguity is a measure commonly used in geospatial statistics, where each model unit is considered to be a neighbor if it shares any boundary (edges or corners) with another model unit of the same layer. This method allows for a more inclusive neighborhood structure than 'Rook' contiguity, which only considers edge-sharing neighbors.

**Application to fMRI vertices.**  We re-analyzed fMRI data from the left hemisphere of 22 sighted subjects from Hauptman et al. (2024). The functional and structural data were pre-processed by Hauptman et al. (2024) and provided as GIfTI files. These files contain the cortical surface representation with 32,492 vertices (GIfTI format, Human Connectome Project benchmark) spanning the left hemisphere. To quantify the spatial autocorrelation in the fMRI data (fig. 3), we define $w_{ij}$ such that each vertex is considered to have six direct neighbors, reflecting the surface mesh structure where each vertex connects to six surrounding vertices based on their adjacency on the cortical surface.

Independent of the input modality (model layer units, fMRI vertices) Moran's $I$ was always applied to un-thresholded maps. This is because thresholding a map before applying Moran's $I$ would artificially inflate the estimate of spatial autocorrelation, as it results in contiguous patches of $0$ values.

## B  HAUPTMAN ET AL. (2024): DESIGN AND FMRI PROCESSING

**fMRI Processing.**  Both Elli et al. (2019) and Hauptman et al. (2024) are based on fMRI recordings in response to the same task and stimuli—similarity judgments of auditorily presented words—and both investigate the relationship between sensory input and conceptual representations in the human cortex. Elli et al. (2019) ($n = 13$) provide evidence suggesting that lexico-semantic information about verbs and nouns is represented in partially non-overlapping cortical networks (Figure 1A in Elli et al. (2019)). The data from Hauptman et al. (2024) are available at OPENICPSR.

Hauptman et al. (2024) presented the same auditory stimuli to 21 congenitally blind and 22 sighted subjects and investigated how neural representations of sensory concepts develop independent of sensory experience; here, we only include data from the 22 sighted subjects in our analysis. Similarly to Elli et al. (2019), Hauptman et al. (2024) find distinct verb- and noun-selective regions (e.g. Figure 3A in Hauptman et al. (2024); Figure 7 below). For scanning parameters, and details on the fMRI preprocessing pipeline, see Hauptman et al. (2024).

**Experimental Design.**  In each 2-second trial, subjects assessed the similarity of two auditorily presented words from the same verb or noun subcategory (e.g., bird or mammal for nouns, light

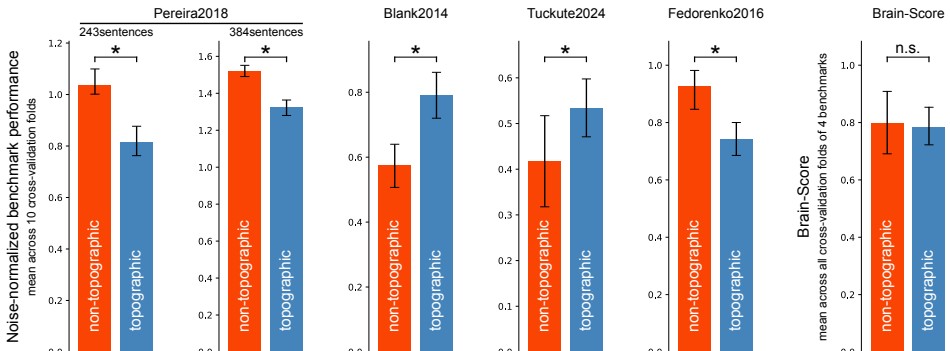

Figure 5: **Brain-Score brain alignment performance (linear predictivity).** We tested TopoLM and the non-topographic control on Brain-Score language using linear predictivity to estimate alignment. TopoLM outperforms the control in Blank2014 and Tuckete2024, but performs slightly worse at Pereira2018 and Fedorenko2016 ($t$-test performed for each benchmark separately: $p < 0.05$). For each benchmark, we sampled from 10 cross-validation loops to compute the bootstrapped 95% confidence intervals (black bars). When results are averaged across the 4 benchmarks (last panel 'Brain-Score'), we don't find evidence for a significant difference between TopoLM and its control ($t$-test: $p > 0.05$).

or sound for verbs). Four trials, all using stimuli from the same subcategory, form a mini-block, followed by a 10-second baseline condition. This yielded a total of 288 pairs of nouns and 288 pairs of verbs[9].

To evaluate each vertex's selectivity for a mini-block, we ran a general linear model with one predictor per mini-block. Contrasting verb and noun mini-blocks produced a $t$-value for each vertex, indicating selectivity toward verbs vs. nouns. As in Hauptman et al. (2024), we based our analysis on data from the left hemisphere, which is more involved in language processing (Frost, 1999; Szaflarski et al., 2006); this is standard in work on language neuroscience.

## C  MOSELEY & PULVERMÜLLER (2014): EXPERIMENTAL DESIGN.

In each trial, 18 participants from Moseley & Pulvermüller (2014) silently read a single word from one of four categories: abstract nouns (e.g. *clue*, *guide*), concrete nouns (e.g. *rice*, *goose*), abstract verbs (e.g. *heal*, *dwell*), and concrete verbs (e.g. *knit*, *poke*). In total, there were 40 stimuli in each category for a total of 160 words. For additional information on the experimental design and neuroimaging parameters, please see Moseley & Pulvermüller (2014).

## D  BRAIN-SCORE LANGUAGE DATASETS AND PERFORMANCE

**Blank et al. (2014)**  This dataset consists of fMRI signals recorded from 12 functional regions of interest (fROIs), offering a lower resolution than the dataset used by Pereira et al. (2018). Five participants listened to eight naturalistic stories, adapted from fairy tales and short stories (Futrell et al., 2018). Each story lasted approximately five minutes, containing around 165 sentences, providing significantly longer context compared to other neuroimaging datasets.

**Fedorenko et al. (2016)**  This dataset captures ECoG signals from five participants as they read sentences consisting of eight words, presented one word at a time for either 450 or 700 ms. Following Schrimpf et al. (2021), we focus on 52 out of 80 sentences presented to all participants.

---

[9]Each pair was presented with verbs in the infinitive (with 'to') and nouns in the definite (with 'the'). Removing these determiners when feeding stimuli through the model results in visually more distinct selectivity clusters; see Figure 10.

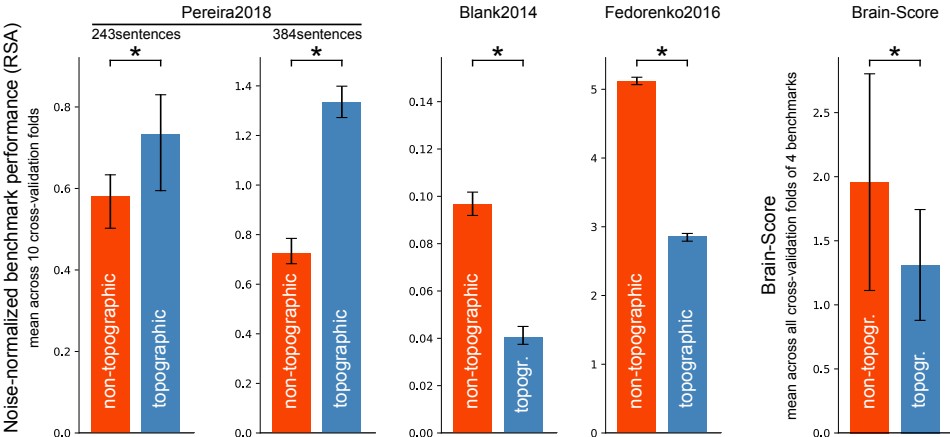

Figure 6: **Brain-Score brain alignment performance (RSA).** We tested TopoLM and the non-topographic control on Brain-Score language using representational similarity analysis (RSA) to estimate alignment. TopoLM outperforms the control in Pereira2018, but performs worse at Blank2014 and Fedorenko2016 ($t$-test performed for each benchmark separately: $p < 0.05$). For each benchmark, we sampled from 10 cross-validation loops to compute the bootstrapped 95% confidence intervals (black bars). When results are averaged across the 3 benchmarks (last panel 'Brain-Score'), we find evidence for a significant difference between TopoLM and its control ($t$-test: $p > 0.05$). However, this result needs to be interpreted with caution as it is dominated by the results from Fedorenko2016 which only show very large absolute values, because the underlying data used for normalization is very noisy, i.e. the noise ceiling is very low.

**Pereira et al. (2018)**  This dataset contains fMRI activations (BOLD responses) recorded as participants read short passages, with each sentence displayed for four seconds. It consists of two experiments: the first with nine participants reading 384 sentences, and the second with six participants reading 243 sentences. Each experiment covered 24 topics. Results are reported as the average alignment across both experiments, normalized using cross-subject consistency estimates.

**Tuckute et al. (2024)**  This dataset is used to assess the distributional robustness of language models. Five participants read 1,000 six-word sentences, each presented for two seconds. BOLD responses from language network voxels were averaged both within and across participants, providing an overall response for each sentence. The stimuli span a wide linguistic range, allowing for diverse model-brain comparisons. Randomized sentence order and averaging across participants reduce temporal autocorrelation effects, making this dataset particularly challenging for model evaluation due to the linguistic diversity.

# E  SUPPLEMENTARY FIGURES

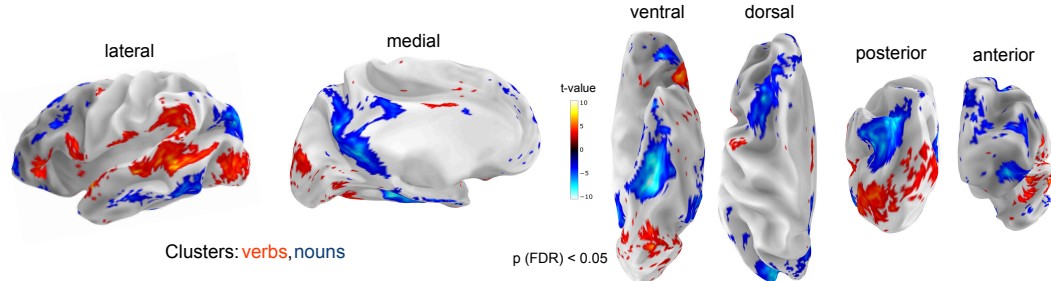

Figure 7: **Verb/noun-clusters across the human cortex.** Re-analyzing data from Hauptman et al. (2024), we replicated their main findings in that we found verb-/noun-selective clusters (red vs. blue) across the left hemisphere. We here show group-level results of all sighted subjects at $p(\text{FDR}) < 0.05$.

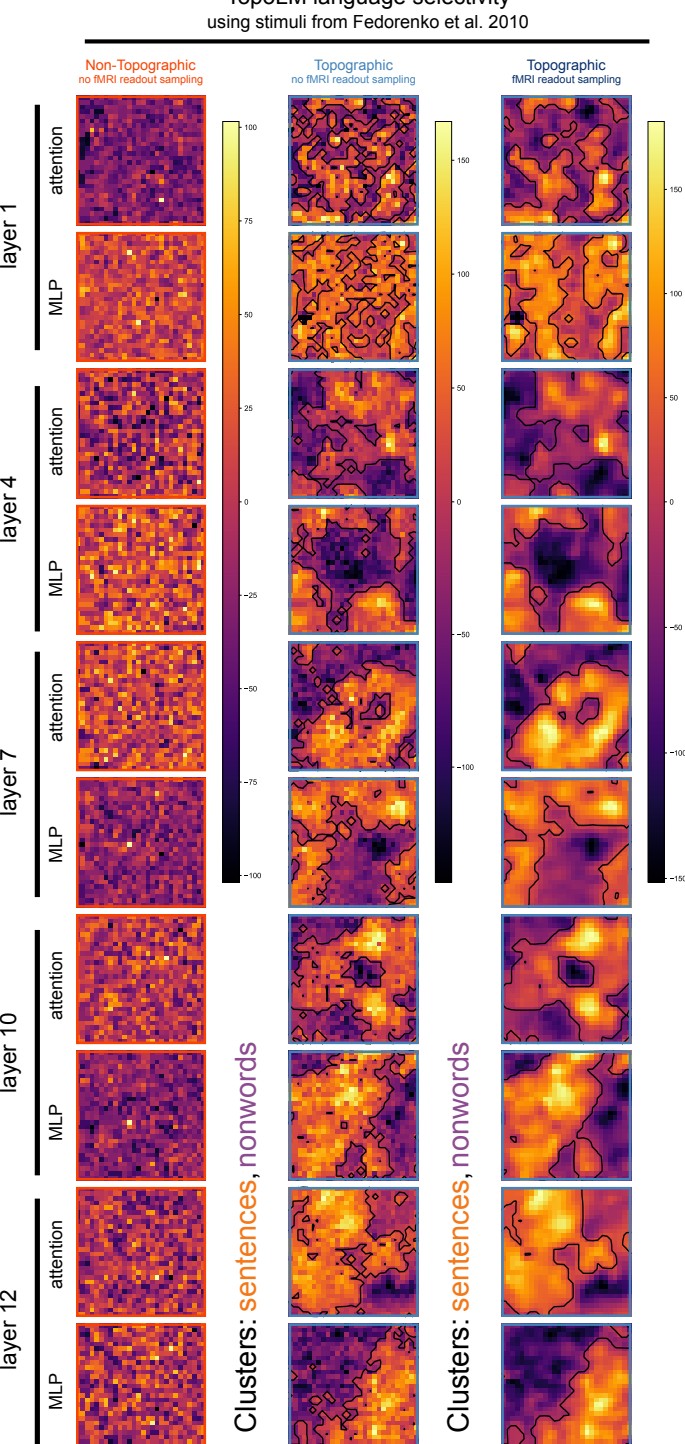

Figure 8: **Core language system.** Using stimuli from Fedorenko et al. (2010), we create contrast maps reflecting language-selectivity for each unit of a given layer (language-selective cluster in yellow). We here show maps for TopoLM with (right column, dark blue) or without (center column, light blue) simulating the fMRI readout sampling process. Using a cluster growing algorithm we identify language-selective clusters. For comparison, we also show the language-selectivity maps from the non-topographic control (left column, orange frames).

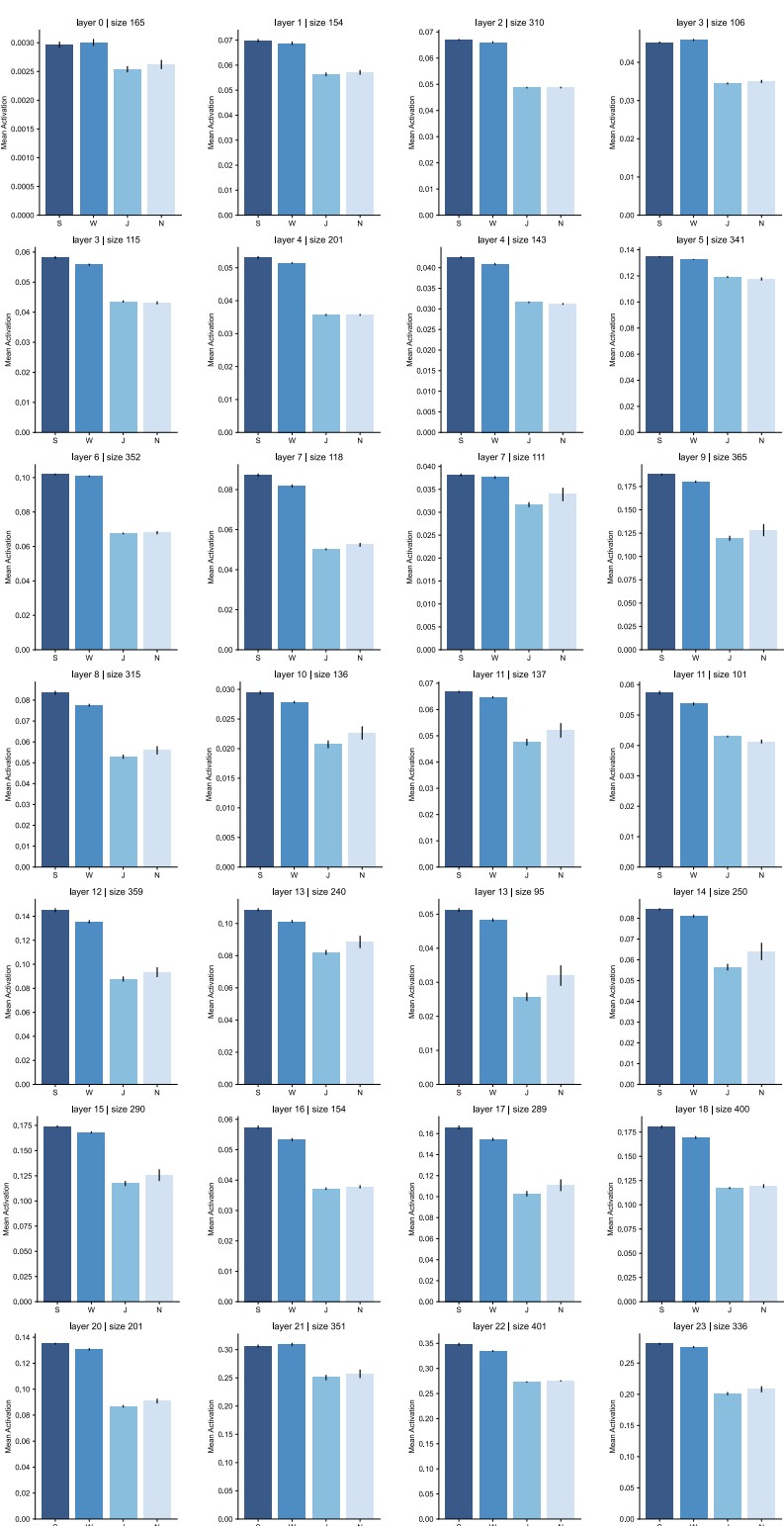

Figure 9: **Consistent response profiles across language-selective clusters in TopoLM.** Using stimuli from Fedorenko et al. (2024), we extract response profiles to 4 linguistic categories across language-selective clusters. Importantly, the observed response profiles are consistent across language-selective clusters. For brevity, here, we only show clusters with more than 100 units.

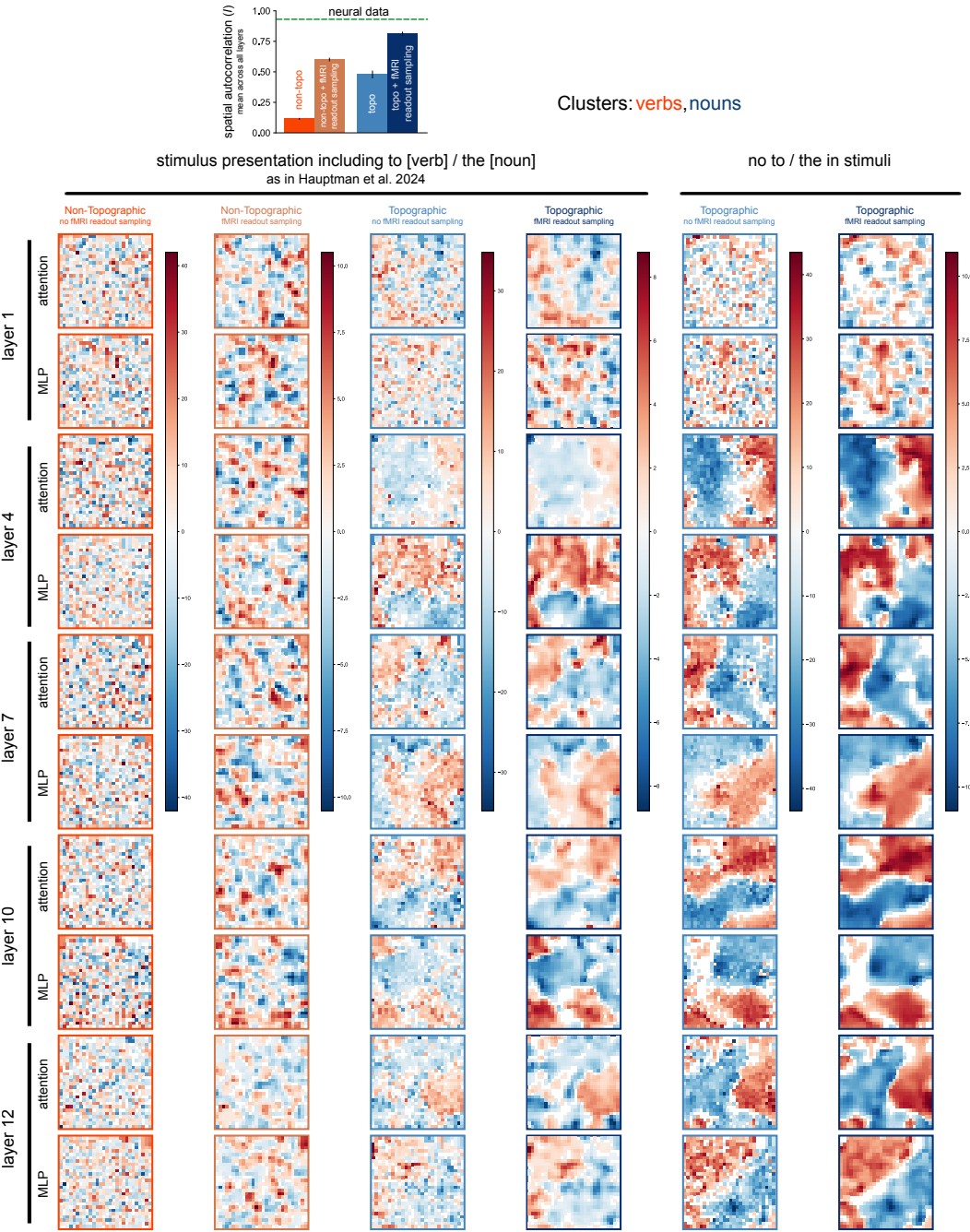

Figure 10: **Verb-noun clusters across TopoLM layers in response to stimuli from Hauptman et al. 2024.** Complementing Figure 3, here, we show autocorrelation estimates (Moran's $I$) for both TopoLM and its non-topographic control with and without FWHM filtering to simulate the fMRI sampling process (upper panel) and the associated verb-noun contrast maps (verb: red / noun: blue) for 5 layers (columns 1-4). When the stimuli are presented without 'to' or 'the' to accompany verbs and nouns, respectively, the verb-noun clustering becomes even more prominent (columns 5-6).

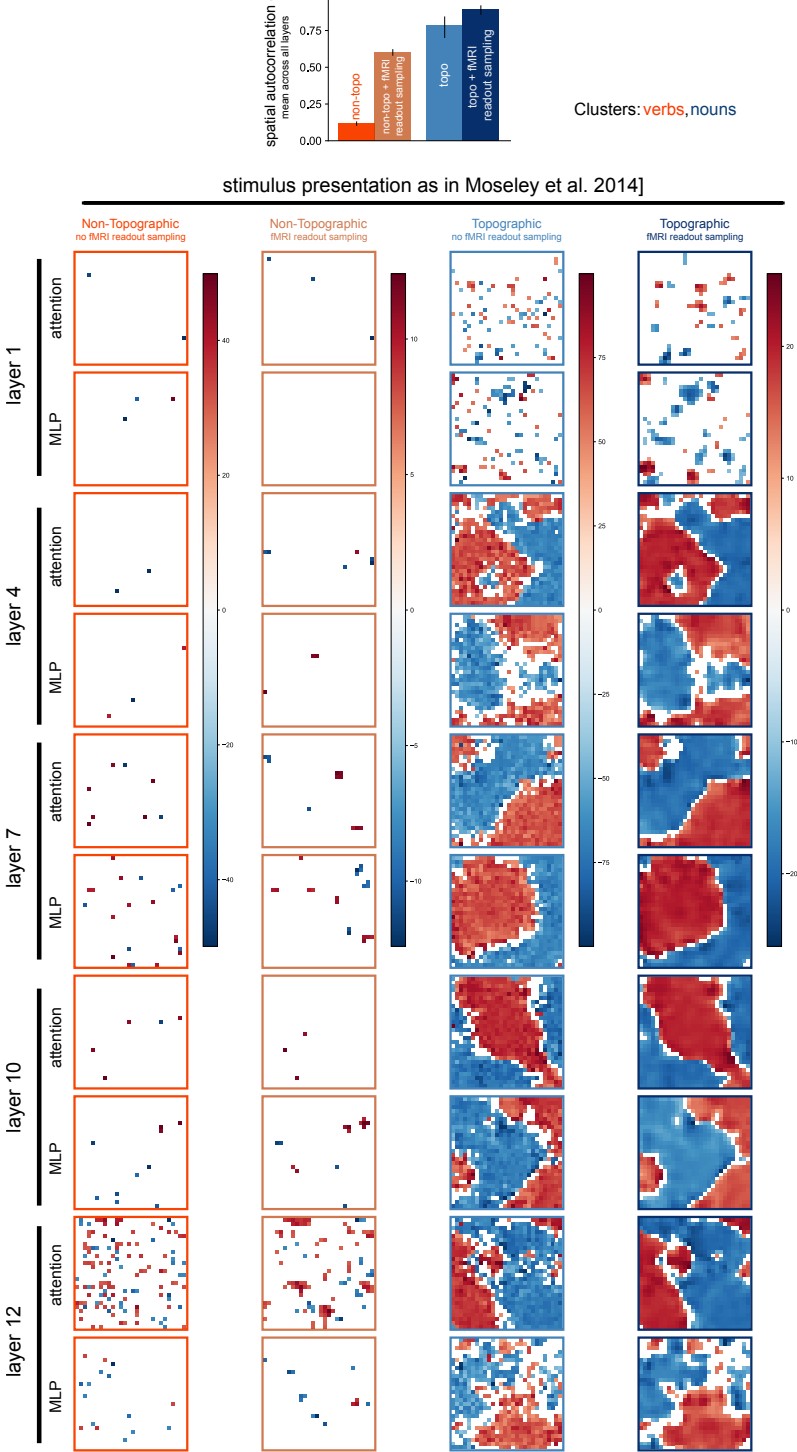

Figure 11: **Verb-noun clusters across TopoLM layers in response to stimuli from Moseley et al. 2014.** Complementing Figure 4, here, we show autocorrelation estimates (Moran's $I$) for both TopoLM and its non-topographic control with and without FWHM filtering to simulate the fMRI sampling process (upper panel) and the associated verb-noun contrast maps for 5 layers. Autocorrelation estimates are based on the verb-noun contrast maps (verb: red / noun: blue).

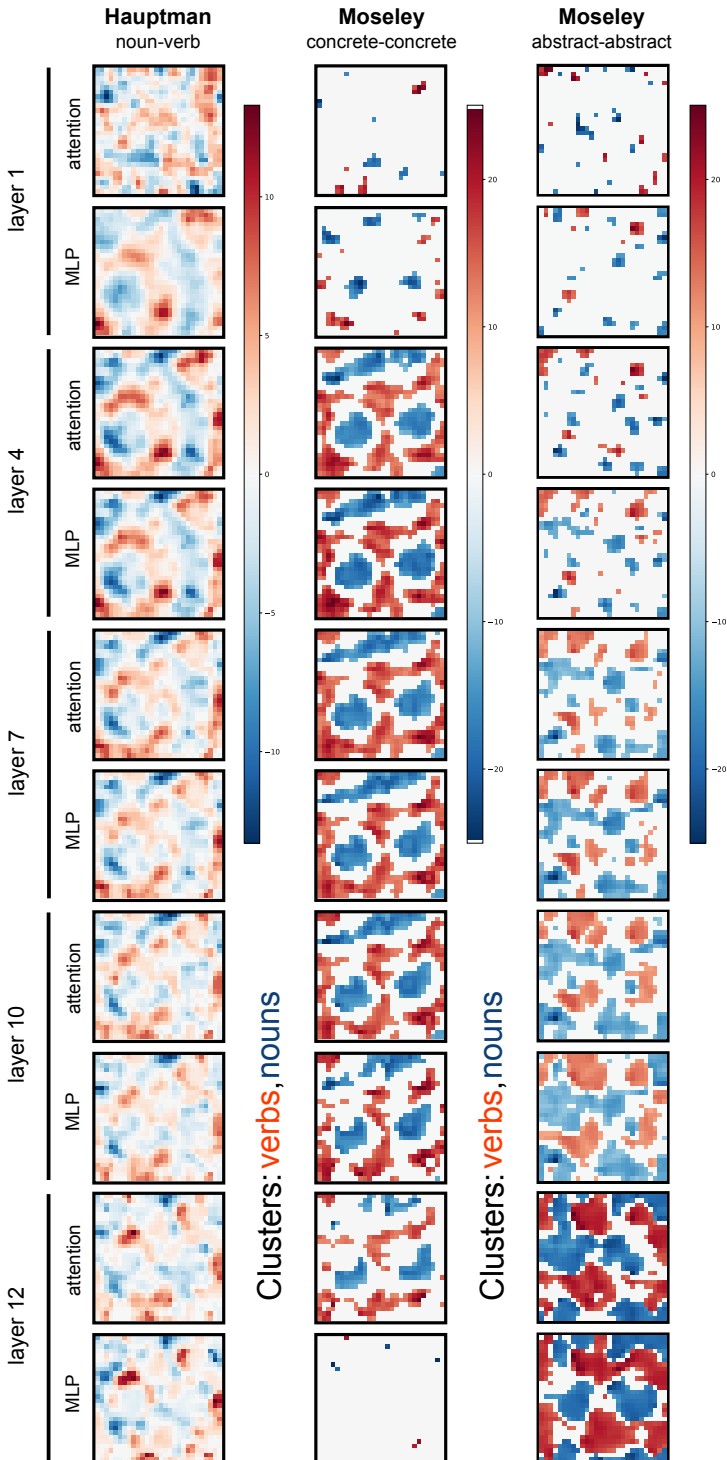

Figure 12: **Verb-noun clusters in a TopoLM without randomly permuted positions.** As described in Footnote 2, we trained an equivalent model without first randomly permuting the spatial encoding of each layer. We evaluate this model on the stimuli from Hauptman et al. (2024) and Moseley & Pulvermüller (2014). We see that the spatial activation pattern propagates through the entire network, as the model no longer has to learn unique spatial maps for each layer in order to minimize spatial loss. Verb and noun clusters shown in red and blue respectively.

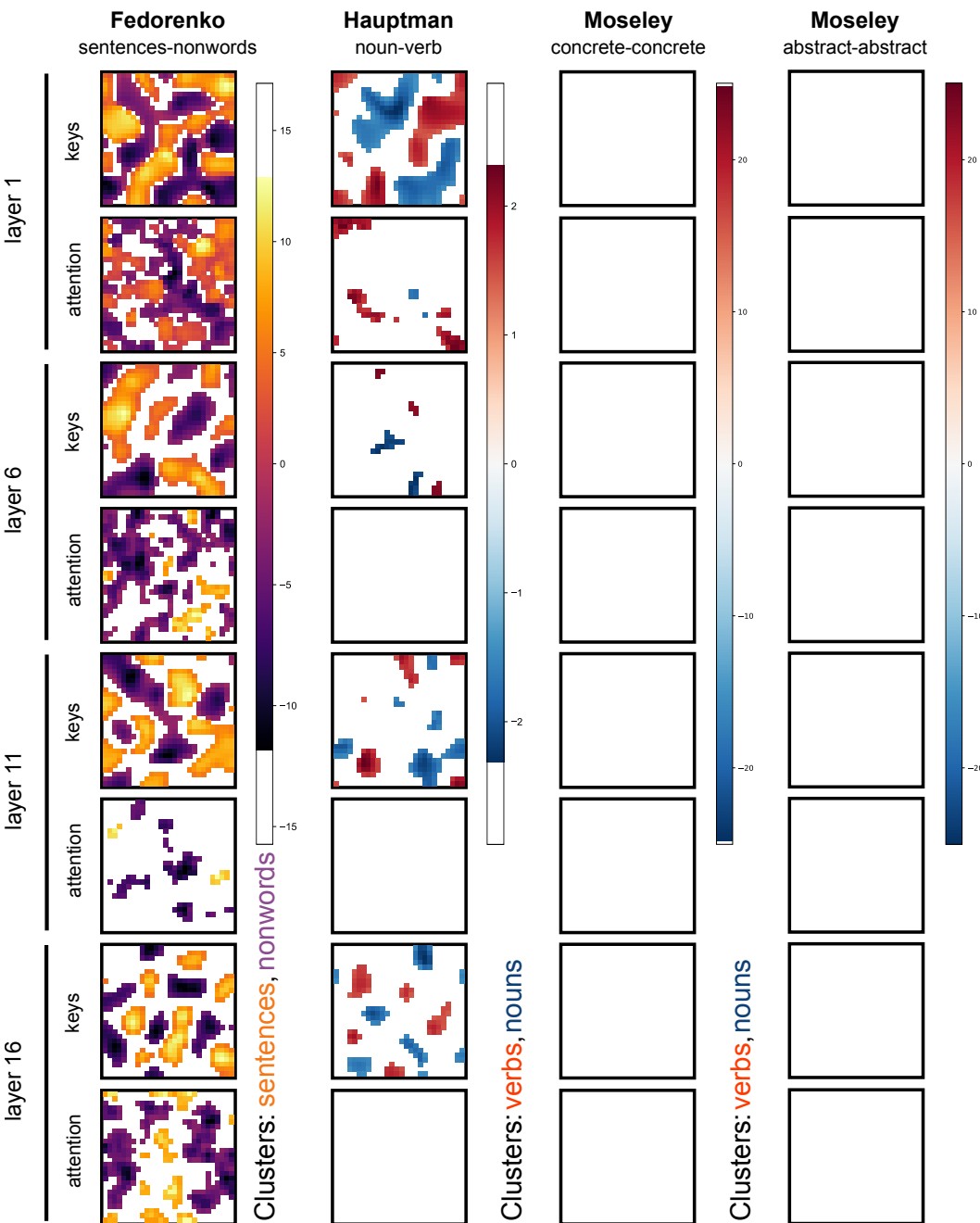

Figure 13: **Topoformer-BERT Evaluations.** Contrast maps (with fMRI-like readout sampling) for Topoformer-BERT (BinHuraib et al., 2024) on stimuli from Fedorenko et al. (2010), Hauptman et al. (2024), and Moseley & Pulvermüller (2014). We plot contrasted activations after multiplication by the key matrix and after the entire attention layer, as this is where local connectivity is applied. We note that few units come out significant for the stimuli from Hauptman et al. (2024) and Moseley & Pulvermüller (2014); however, we identify some clusters in Fedorenko et al. (2010)'s language localization task.

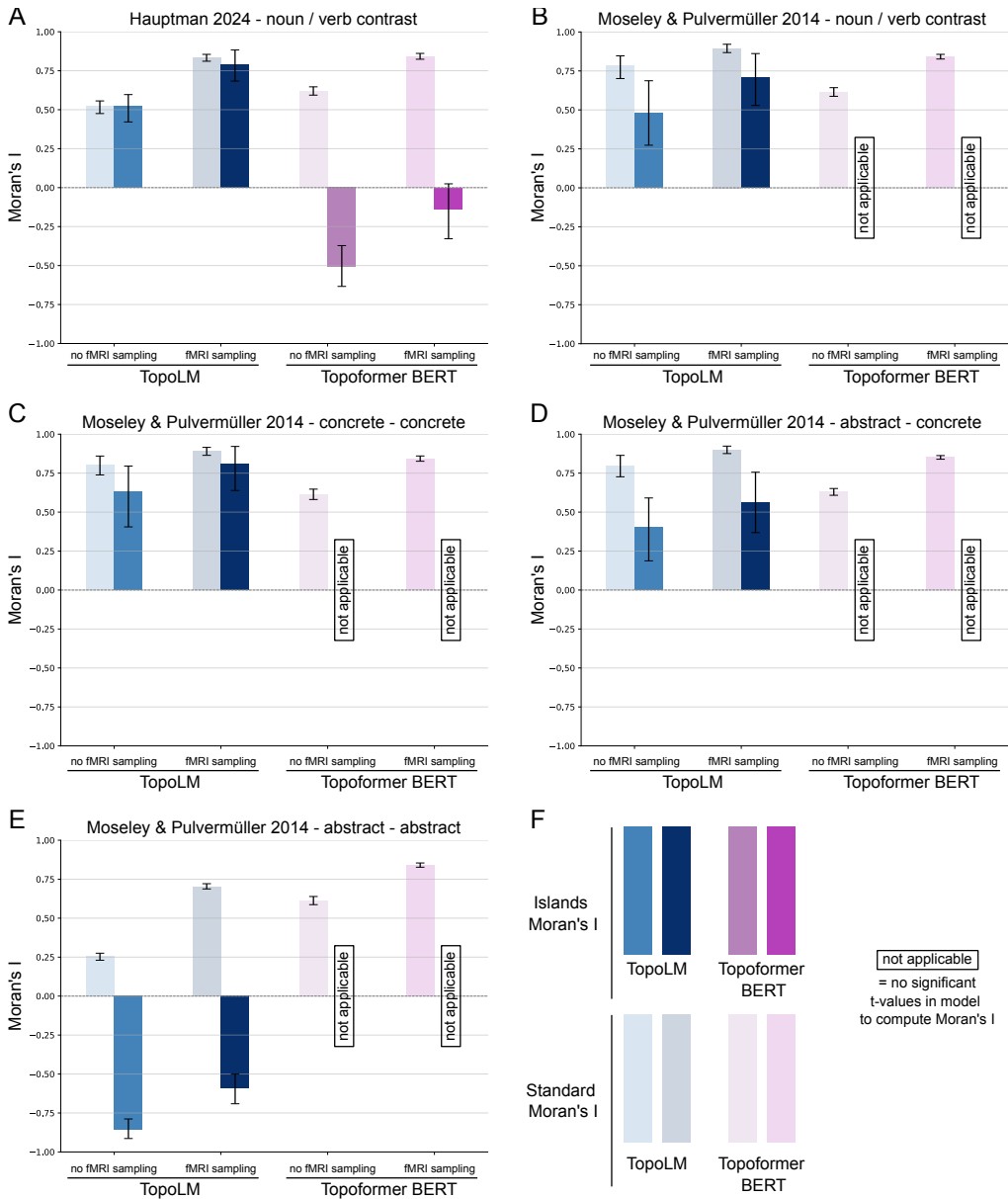

Figure 14: **TopoLM comparison of clustering with baseline model.** We compared TopoLM to an additional baseline model (Topoformer BERT, BinHuraib et al. (2024)) with regard to clustering. Most t-values across contrasts in multiple datasets (Hauptman et al. (2024); Moseley & Pulvermüller (2014)) are not significant in Topoformer BERT (after applying FDR correction as has been done in TopoLM). Thus, we estimated autocorrelation using a variant of Moran's I only considering "islands" of units with significant t-values for a given contrast (and averaging across islands while not considering island size to obtain a single estimate per map). We computed "Islands Moran's I" for both TopoLM and Topoformer BERT. **A)** Using this metric, we find strong clustering in TopoLM (Island's Moran's I = 0.52 without simulated fMRI sampling, and Island's Moran's I = 0.83 with sampling), numbers matching our results when computing Standard Moran's I (I = 0.53 without simulated fMRI sampling, I = 0.85 with sampling). In contrast, we observe patterns of dispersion or neither clear dispersion, nor autocorrelation in Topoformer BERT (Island's Moran's I = -0.51 without simulated fMRI sampling, and Island's Moran's I = -0.14 with sampling). **B-E)** With the exception of one contrast (see panel E), TopoLM results based on data from Moseley & Pulvermüller (2014) using Islands Moran's I qualitatively match the results obtained by using Standard Moran's I. In contrast, in Topoformer BERT autocorrelation using Islands Moran's I cannot be computed for the contrasts in Moseley & Pulvermüller (2014) as t-values underlying the contrast maps are not significant. **F)** Legend.

