# OpenReview forum: "TopoLM: brain-like spatio-functional organization in a topographic language model"
_ICLR.cc/2025/Conference — ICLR 2025 Oral_

### Official Review · Reviewer_U3qv · 2024-10-27

**Soundness:** 3
**Presentation:** 3
**Contribution:** 3
**Rating:** 8
**Confidence:** 4

**Summary:**

The paper presents a language model whose training objective is based on a joint loss function: the standard next-word prediction loss and an additional spatial regularization term, which enforces a constraint where adjacent neurons exhibit more similar activation profiles than those of more distant neurons. The primary contributions of the paper are: (1) demonstrating that such a model can be successfully trained, resulting in an increase in local spatial correlations, (2) showing that its performance on standard benchmarks is comparable to models trained without the spatial regularization, and (3) providing evidence that the learned representations exhibit some degree of isomorphism with neural representations. For example, certain spatial clusters show stronger activation in response to well-formed sentences compared to nonsensical ones, and there appears to be a distinction between cluster-responses to nouns and verbs.

**Strengths:**

The paper addresses a novel and important issue. There has been little work on training LLMs with topography constraints.  Successful training opens the door to interesting analyses, and potentially more flexible approaches for supervised learning of model-to-brain alignment. The writing is clear, and the analyses are interesting in that the authors apply well established statistical approaches used in the human neuroimaging literature to the model itself.

**Weaknesses:**

**P.S  following the discussion period: I believe the more critical perspective that was adopted by the authors during the review process improves the soundness of the paper. I have modified my evaluation.**

1. The spatial loss term is a key element in the work and should be better motivated.
1a. As phrased in the paper, the loss term constrains neurons that are closer to each other (according to a predefined spatial arrangement) to have more strongly  correlated activation profiles than those farther apart. However, this introduces a global, long-range constraint, because activation profiles of neurons that are far apart are not independent, but constrained to differ. To enforce a strictly local connectivity constraint, the authors could have employed a formulation that emulates Gaussian smoothing or another local averaging kernel, which would only consider the immediate neighborhood of each neuron. Reading the authors' motivation for the spatial regularization term on page 3, I expected a local spatial smoothing constraint rather than the more global constraint that was implemented. They write: "This loss function measures spatial smoothness, which serves as an efficiently computable proxy for neural wiring length: neurons located close to one another should have similar response profiles—i.e., their activations should be correlated (Lee et al., 2020)". In practice, for computational efficiency, the authors do not transform an entire layer to a grid-topology, but select 5 smaller areas; however, the point still holds.

1.b In addition, the spatial-loss term introduces certain computational characteristics that should be considered in greater depth. Given its properties, it may encourage learning of decorrelated features, which could be advantageous for downstream transfer learning and fine-tuning.  In fact, this principle is used in a regularization technique introduced by Zhu et al (2023; https://arxiv.org/abs/2306.13292) to learn a potentially richer feature set. On the other hand, the spatial regularization term also pushes in the opposite direction: it enforces strong correlations between nearby units, potentially reducing the model’s effective capacity. In the extreme case, it can force two adjacent units to have almost identical activation patterns. This could limit the model’s ability to learn distinct features, which is a fundamental goal in neural network training. The authors briefly touch on this issue in the results section (p. 9), where they mention that the spatial regularization may help prevent overfitting, but this point asks for a more thorough treatment.

2. The effects of the spatial loss on neuron-to-neuron similarity should be quantitatively evaluated
It is not clear how and whether TopoLM training changes the covariance structure of neural firing. It is important to rule out the possibility that non topographic models produce similar unit-to-unit correlation patterns compared to TopoLM. Otherwise, it would mean that training with spatial loss is not necessary for obtaining the resulting covariance structure.  This could be done by computing, for each neuron, a histogram of correlation values and averaging these histograms across all neurons.  If both TopoLM and standard models produce similar correlation distributions, it would suggest that typical neural networks naturally develop a graded connectivity structure similar to what TopoLM enforces, but this structure may manifest in a shuffled form in the absence of an explicit spatial arrangement.

3. Some of the spatial results appear determined or constrained by the training objective and other procedures.
The finding that training produces stronger spatial smoothness and clustering seems to me to be a direct consequence of the spatial constraint in the training objective, which inherently encourages the formation of locally correlated clusters with weak correlations between remote clusters. Essentially, it implements a result similar to spatial data smoothing. Similarly, the fact that a simulation of fMRI readout-sampling (which applies further local averaging of unit activations) produces an even smoother data map is expected, as this is a direct result of the averaging imposed by the readout kernel.  Consequently, these properties do not emerge organically from the model’s spatial-loss training or fmri-like sampling, but instead reflect the successful implementation of the externally imposed constraints and workflow.  Because of this, the documentation of spatial clusters in statistical analyses is not surprising. For example, the authors write (p. 7) that “Contrasting model activations to verb and noun stimuli yields clear verb- and noun-selective clusters across layers in our topographic model”, but given the smoothing constraint applied, it would be very likely that the results would manifest in the form of clusters, as nearby units are constrained to be non-independent.  If these results are rephrased to be presented as a validation or sanity check that would make sense, but currently they are phrased as if they are a novel result.

4. Some of the language-activation results appear dependent on the definition of language regions
The authors define a language system in TopoLM as the set of units that respond more strongly to grammatically valid sentences than to strings of non-words, which is analogous to the use of a functional localizer task in fMRI studies.  Then, for the network units identified by this condition, they evaluate responses to well-formed sentences and different types of language and language-like stimuli including strings of scrambled non-words or stings of phonotactically plausible non-words (jabberwocky sentences).  However, the aforementioned test used to define the localizer is not entirely independent of the subsequent evaluations. While it does not fully determine them, the localizer identifies units that are likely sensitive to word meaning and syntactic relationships, which are properties not found in the other stimulus types. Therefore, it is not entirely surprising that in the areas identified by the localizer, the strongest responses were to sentences, as compared to unconnected words, non-words, or Jabberwocky sentences.

It would have been more convincing if these results were found without the use of a localizer, using the spatial distribution alone: if the local clusters formed in the network are functionally meaningful, it would be important to see what information is encoded in those clusters.  If the strongest clusters naturally show a gradient of responses (e.g., from sentences to unconnected non-words), this would suggest that such distinctions are inherently emphasized during learning. Clusters can be identified via regional homogeneity (as done for fMRI) by computing correlation in response profiles across neurons.

The last two points also bear on the author’s ultimate conclusion, where they write, “Our results suggest that the functional organization of the human language system is driven by a unified spatial objective..”  I don’t think this follows from the results,  because I don’t see a basis for the analogy between the explicit spatial objective implemented in the paper (which produces some of the results) and those factors that ultimately lead to local correlations in the brain. Specifically, in the brain it is probably *unlikely* that there exists an objective that constraints nearby areas to fire similarly. This more likely emerges from cytoarchitecture constraints which produce different gradients of correlation (sometimes within the same neural population; https://pmc.ncbi.nlm.nih.gov/articles/PMC3412441/).

5. General comment: I do not see a general weakness in the fact that benchmarks are not consistently improved by TopoLM. But what is lacking are more clear statements about whether TopLM develops different representations than non-Topo architectures.  This can be quantified by comparing representation-dissimilarity matrices, CCA, CKA or other methods for quantifying similarity of representations.  Of course, a comparison of these against RDMs computed from brain areas could also be helpful.

**Questions:**

1.  I did not understand the argument about smoothing reducing wiring length. From my understanding, the TDANN network, even after training with the spatial regularization, retains the same dimensionality (weight matrix) as a non-spatial network. What then is the basis for the claim that this minimizes wiring length? (This refers to p. 3: “The TDANN’s spatial smoothness, as implemented with an additional loss term, is an indirect but efficient approach to minimizing local wiring-length.”) Additionally, from a functional perspective, in vision, hierarchical processing may reduce the need for long-range connectivity, as much processing occurs in localized circuits (even though reciprocal connections still exist). But in language processing, long-range connectivity seems essential. For instance, world knowledge can immediately influence lexical processing, and predictions can impact activity in primary and secondary auditory cortices, indicating that long-range connections are necessary in this domain.

2. page 6, the statement that a successful alignment between brain and model is accompanied by “these clusters have consistent response profiles”; the meaning of this phrase was unclear.

3. The analysis is focusing on univariate methods, where preference for sentence or word class is identified by differences in single-unit activation. But it is well known that the brain also represents information using multivariate (population) codes. What if a set of units, each of which shows similar activation for two classes of linguistic stimuli, can still be used to decode stimulus class through a population code?  From the perspective of the current work, these units will be defined as being outside the identified ‘selectivity’ clusters (and hence not considered ‘core’ to language). However, to the extent that a linear probe indicates that this population-code contains relevant information, there is no a priori reason to exclude them as being less relevant, essential or core to language.  For example, the null effect reported for sentences vs. unconnected words (p.6; results section) could still be consistent with the possibility that this set of units can differentiate between the two classes if a classifier trained and tested on their activations using linear probing. In short, the authors should better motivate why the operationalize ‘centrality’ to language based on univariate tests rather than population codes.

4. For the benchmarks (p. 9) it seemed that fine-tuning involved retraining the entire model, rather than freezing the trained model, using it as feature extractor, and using a linear head for classification.  To evaluate the quality of features produced (rather than the model’s capacity to learn a new task) it would be useful to treat TopoLM and the control architecture as feature extractors, and apply the same procedure for learning a new task.

References

Zhu, J., Evtimova, K., Chen, Y., Shwartz-Ziv, R., & LeCun, Y. (2023). Variance-covariance regularization improves representation learning. arXiv preprint arXiv:2306.13292.

---

> ### Author Response · Authors · 2024-11-24
> **Response to questions**
>
> Thank you very much for your thorough and thoughtful comments and suggestions. One of your main suggestions is to compare TopoLM to an additional topographic baseline language model. We are currently testing Topoformer BERT (BinHuraib 2024, https://openreview.net/forum?id=R6AA1NZhLd) on all three main neural datasets we present in the paper (Fedorenko 2024, Hauptman 2024, Moseley 2014) and will post the results and an updated version of the paper as soon as possible.
>
> In your first question you are asking about the relation between the spatial correlation loss and reduced wiring length. For a detailed answer to this point, please see our response below to weakness 4, paragraph 3 that you raised.
>
> Your second question rightfully asks to clarify the second condition under which we see the spatio-functional alignment between brain and model to be successful. We hope to make this point more concrete by adding the following footnote to the sentence in question: ”The language areas all show a similar response profile (despite slight apparent differences, no region by condition interactions come out as reliable, even in well-powered studies.” Fedorenko et al. (2024b). In other words, as Fedorenko et al. 2024b find similar differences in response profiles across subregions of the core human language system, we consider similar response profiles across subregions in the model as a sign of good model brain alignment.
>
> Your third question addresses the type of statistical analysis (univariate analysis) of the Fedorenko et al. 2024b data. We agree that information in the human (visual) cortex is represented in multivariate population codes (Kriegeskorte 2008, https://doi.org/10.3389/neuro.06.004.2008; Kriegeskorte 2011, https://doi.org/10.7551/mitpress/8404.001.0001). Decoding and other multivariate pattern analyses can reveal information in regions in which univariate approaches might be less sensitive or even fail to provide evidence for a given hypothesis. For example, Nakuci 2024 (https://doi.org/10.1162/imag_a_00359) recently provided evidence suggesting that “behavioral signatures can be decoded from a much broader range of cortical areas than previously recognized”. While these are important considerations, we focused on describing basic functional organization (core language system, verb/noun clusters etc.) to which more sophisticated and potentially more sensitive tools from the multivariate toolbox can very likely also provide answers, but might not be necessary. In future investigations of TopoLM one might want to focus on finer differential model responses not detectable by the type of univariate analyses we present here.
>
> Thank you for your question 4 pertaining to the way in which downstream behavioral performances were benchmarked. In addition to fine-tuning on the individual test (of which results are already provided), in an additional analysis we will leave all model weights frozen, perform the evaluation without further fine-tuning, and report the results as soon as possible.
>
> Below, we separately respond to the weaknesses you raised.

---

> ### Author Response · Authors · 2024-11-24
> **Response to weaknesses 1-3**
>
> Thank you again for raising these potential weaknesses, most of which (1b, 2, 3) we will respond to in more detail using additional analysis we still need to complete.
>
> Weakness 1a. As you point out, in the current implementation of the loss we sample from local neighborhoods. Even if we were not doing so for computational efficiency reasons, this loss does - to our understanding - not introduce a constraint yielding remote units to have dissimilar response profiles. In contrast, even when sampling from the entire layer, our loss only encourages nearby units to respond similarly while leaving response profiles of remote units as similar or dissimilar as they were when no spatial correlation loss was applied.
>
> Weakness 1b: induced correlation structure within a layer encouraged by the spatial correlation loss term. We will try to address this point with an additional analysis, and will keep you posted in this regard.
>
> Weakness 2: quantitative evaluation of the covariance structure of the response profiles across a layer. We are currently conducting an additional analysis and will respond with the results as soon as possible.
>
> Weakness 3: the formation of clusters is not surprising. We believe we have not demonstrated clearly enough that the emergence of clusters is in our eyes non-trivial. As mentioned above, in an additional analysis, we are currently testing another topographic language model (Topoformer BERT) on the neural datasets we report in the paper and will report the results as soon as possible.

---

> ### Author Response · Authors · 2024-11-24
> **Response to weaknesses 4-5**
>
> Your weakness 4, paragraph 1 addresses the potential recursiveness of the core language system results. Here you suggest the replication of results from Fedorenko et al. 2024b not to be surprising due to a too large similarity between linguistic stimuli used to localize language-selective regions and the stimuli of the main experiment. We agree that a certain similarity between these two ideally independent sets of stimuli cannot be excluded. However, our main goal was to replicate findings in the neuroscience literature for which data is available—in particular, we wanted to highlight the fact that TopoLM allows for the “by-region” response profile analysis typical of much language neuroscience work in a way that non-topographic language models cannot (Fig. 2a, 2b) We are not aware of (readily available) neuroimaging data with larger independence between localizer and main experimental linguistic stimuli, but we would be very happy to use it for model testing if it exists.
>
> Weakness 4, paragraph 2: Clarification of claim on unified spatial objective in the conclusion. Thank you for your suggestion that cytoarchitectural constraints might produce gradients of correlation, and that our spatial correlation loss might not be related to factors ultimately leading to local cortical correlations. Addressing your first point, we agree that cytoarchitectural constraints, such as gradients in receptor density and local connectivity patterns, have an influence of producing gradients of correlations of nearby neurons. The paper you cite (Langers at al. 2011) demonstrates this well in that frequency gradients in the auditory cortex align with clearly identifiable anatomical locations, such as Heschl’s gyrus or the planum temporale which are strongly shaped by cytoarchitectural constraints such as neuron density (also see Galaburda 1980 https://doi.org/10.1002/cne.901900312, Dick 2017, https://doi.org/10.1523/JNEUROSCI.1436-17.2017). However, studies on sensory-deprived individuals (blind or deaf people), suggest that the same gradients as observed in subjects with intact visual or auditory perception fail to emerge without environmental input, and thus despite these cytoarchitectural constraints being present. For example, retinotopically organized cortical regions such as early visual cortex do not show such organization in the congenitally blind, but are instead repurposed to process language (Bedny 2017, http://dx.doi.org/10.1016/j.tics.2017.06.003). Further, Lomber 2011 (http://www.nature.com/doifinder/10.1038/nn.2653) provide causal evidence from congenitally deaf cats suggesting a repurposing of parts of “auditory cortex” for visual processing. Last, retinotopic gradients in human V1 are thought to be shaped by correlations in retinal input (Katz & Shatz 1996, https://doi.org/10.1126/science.274.5290.1133). While this is clearly a non-exhaustive list of the literature on this issue, we see cytoarchitectural constraints and sensory input as non-exclusive factors shaping cortical specialization.
>
> Weakness 4, paragraph 3: Relation between our TDANN spatial correlation loss and factors ultimately leading to local correlations in the brain. There is clear evidence for functional objectives leading to wiring length minimization (Jacobs 1992, https://doi.org/10.1162/jocn.1992.4.4.323; Chklovskii 2002, https://doi.org/10.1016/S0896-6273(02)00679-7). For strong evidence relating wiring length minimization to locally correlated response profiles that in turn shapes the organization of smoothly varying gradients (or maps) across the visual cortex, see Chklovskii 2004 (https://doi.org/10.1146/annurev.neuro.27.070203.144226). Using the TDANN loss in vision models, Margalit 2024 demonstrate that this objective encouraging local correlation leads to lower between-area wiring length (https://doi.org/10.1016/j.neuron.2024.04.018, see fig. 7); similarly, Blauch et. al (2022; https://doi.org/10.1073/pnas.2112566119) demonstrate that a wiring-cost minimization term produces local correlation. While we are unable to perform the same analyses in TopoLM for the current submission, we do not see strong evidence suggesting that the same TDANN loss in TopoLM would not lead to reduced wiring length minimization which in turn contributes to the local correlation of response profiles we report for TopoLM.
>
> As we believe this extended discussion (especially weakness 4, paragraph 2 & 3) may interest the audience of our paper, we will add parts of it in the discussion of the paper.
>
> Weakness 5: comparison between TopoLM and its non-topographic equivalent on the level of representations with regard to brainscore benchmarks. Thank you for suggesting another type of metric for evaluating language models on brainscore language benchmarks. We are currently running the same benchmarks using RSA instead of linear predicitivity and will report the results as soon as possible.

---

### Official Review · Reviewer_TBeb · 2024-11-02

**Soundness:** 4
**Presentation:** 3
**Contribution:** 4
**Rating:** 8
**Confidence:** 4

**Summary:**

The study introduces TopoLM, a model based on the transformer architecture, aimed at exploring the spatial functional organization of human language systems. This model optimizes autoregressive language modeling by incorporating two-dimensional spatial encoding, along with task and spatial loss, which encourages response smoothness among adjacent units. Experiments demonstrate that the spatial functional organization generated by TopoLM closely aligns with the neuronal clusters found in the brain's language system. Although TopoLM scores slightly lower on certain behavioral tasks compared to non-topological baselines, its performance is comparable on other downstream tasks and brain alignment benchmarks. This indicates that TopoLM successfully captures the topological structure involved in language processing when trained on natural text, providing new insights into the functional organization of language systems.

**Strengths:**

TopoLM introduces a novel approach to understanding the spatial organization of language processing in the human brain.  By integrating two-dimensional spatial encoding into a transformer architecture, the model effectively captures the topological structure of language, which has not been extensively explored in existing literature.  This innovative perspective opens new avenues for research in language modeling and cognitive neuroscience.  Meanwhile, this research has significant implications for both artificial intelligence and neuroscience.  By aligning computational models more closely with the biological structures of the brain, TopoLM paves the way for more effective natural language processing applications.  Furthermore, it contributes to a deeper understanding of the neural mechanisms underlying language, potentially informing both theoretical frameworks and practical applications in cognitive science and related fields.

**Weaknesses:**

While the paper presents a promising theoretical framework, the empirical evaluation of TopoLM is somewhat limited.  The experiments primarily focus on a small set of tasks, which may not fully capture the model's robustness across diverse linguistic contexts.  Meanwhile, this paper does not sufficiently explore or compare TopoLM with other existing topological models in language processing.  Including a comparative analysis with relevant models could strengthen the argument for TopoLM's effectiveness and novelty.  Highlighting specific improvements or unique aspects of TopoLM in relation to these models would enhance the discussion around its contributions to the field.

**Questions:**

Could you elaborate on the specific architectural choices made in TopoLM?  For instance, what motivated the selection of particular layers or activation functions, and how do these choices contribute to the model's performance compared to traditional language models?

Can you provide more insight into the training process for TopoLM?  Specifically, what datasets were used, and what were the criteria for their selection?  Additionally, how did you address potential issues like overfitting during training?

In your discussion of the biological implications of TopoLM, could you provide references or data from neuroimaging studies that support the connection between the topological aspects you incorporate and the way the human brain processes language?

---

> ### Author Response · Authors · 2024-11-24
> **Response to questions**
>
> Thank you for your careful evaluation of our TopoLM paper and for providing constructive feedback.
>
> We agree that our paper would benefit from an additional topographic baseline beyond TopoLM’s non-topographic variant (as already presented in the paper). We are currently working on testing Topoformer BERT (BinHuraib 2024, https://openreview.net/forum?id=R6AA1NZhLd) on all main neural datasets we present in the paper (Fedorenko 2024, Hauptman 2024, Moseley 2014) and will post the results and the updated version of the paper here as soon as possible.
>
> You are raising a few interesting questions with regard to the selection criteria for architectural features and the training data. As described in Section 3 (“Model Specification and Training”), we use a 12-layer GPT-2-style architecture, with hidden size 784 and 16 attention heads. Other training hyperparameters were chosen via hyperparameter search (for more details, see footnotes 1 and 2 on page 4). The model was trained on a randomly sampled 10B-token subset of the FineWeb-Edu dataset. For a detailed account on topography we refer to the introduction covering this topic for brains (“introduction” section, paragraph 2 & 3) and in models (“related work” section). Please let us know if there is anything we are missing in these descriptions and we are happy to explain in more detail.

---

### Official Review · Reviewer_3QeR · 2024-11-03

**Soundness:** 3
**Presentation:** 4
**Contribution:** 2
**Rating:** 8
**Confidence:** 3

**Summary:**

This work introduces TopoLM, a transformer language model incorporating information about the spatial arrangement of the units to model brain-like activity in the processing of language. In particular, the spatial information is integrated by adding a spatial smoothing term to the classical autoregressive loss of decoder-only transformers. By performing functional localization experiments, the authors identify clusters of artificial neurons in TopoLM whose activations are language-selective. The authors highlight the similarity between TopoLM clusters of activations and response in the brain by conducting experiments on verb-noun and concrete-abstract selectivity.

**Strengths:**

The authors contextualized and described with clarity the strategy introduced to incorporate spatial information of activations in transformers for natural language processing. Both the training details and experiments are very well explained and organized.

The work, inspired by previous efforts in vision models, introduces for the first time (to the reviewer knowledge) explicit spatial information in the loss of transformers trained on natural language.

**Weaknesses:**

Given the strong inspiration from human neural activity studies of the work, the work would benefit from more extended discussion and analyses related to some architectural choices (e.g. random permutation of spatial position between layers and role of multi-head attention).

**Questions:**

Some further experiments and discussions could further strengthen the work:
- How does the choice of randomly permuting positions at each layer influence the representations? It would be extremely interesting to consider different types of mapping between spatial positions at successive layers.
- The authors stress the role of considering architectures with multi-head self-attention. Is this type of architecture expected to better model human neural activity?
- In the NLP community, it has been observed a hierarchical organization of syntactic and semantic information in hidden layers of deep learning models (e.g. Belinkov, ACL, 2017; Manning, PNAS, 2019). Do the TopoLM clusters of activation exhibit some level of hierarchy through the layers of the model?
- In case it is feasible, comparing TopoLM representations/clusters also to Topoformer would further strengthen the quality of experiments.

---

> ### Author Response · Authors · 2024-11-24
> **Response to questions**
>
> Thank you for your evaluation of TopoLM, and for suggesting points we can improve on. We agree that some architectural choices deserve a more extensive discussion and in some cases even additional analyses. We address this in the following way.
>
> We will explain the effect of random permutation of spatial position between layers with an additional analysis using a variant of TopoLM in which this random permutation has not been applied. As soon as this analysis is completed, we will describe it here.
>
> The question of whether multi-head attention, as opposed to single-head attention, is better suited for modeling human neural activity is both intriguing and important. AlKhamissi et al. (2024, https://doi.org/10.48550/arXiv.2406.15109) demonstrated that increasing the number of attention heads in untrained models enhances the predictability of model units with activity in the human language network. Furthermore, language models typically employ multiple attention heads to capture diverse functionalities. To remain consistent with this conventional design, we chose multi-head attention over single-head attention in our approach.
>
> We are currently still testing another topographic language model (Topoformer BERT; BinHuraib 2024, https://openreview.net/forum?id=R6AA1NZhLd) on the main neural datasets presented in our paper (Fedorenko 2024, Hauptman 2024, Moseley 2014), the results of which will be posted here and added to the paper as soon as possible.

---

> ### Author Response · Authors · 2024-11-26
> **Response to question about random permutation**
>
> Regarding Q1: our initial intuition for randomization was to more closely mimic the brain by minimizing the extent to which the model could utilize the feed-forward nature of Transformers—in particular, if positional embeddings were equivalent across layers, the model would learn no variance in topography between layers, since the topographic structure could just propagate through the network. While the brain does have feed-forward connectivity, this hierarchical organization is very different from that of a language model; thus, our goal was to abstract away from this as much as possible by using random spatial embeddings.
>
> To empirically test this, we ran a small-scale analysis of the effect of randomizing the spatial embedding bijection. We trained a model without randomized spatial embedding for 110 iterations (all other hyperparameters remain the same as the main model presented in the paper). We note that the results are in line with our intuitions: the model exploits the fact that activations feed forward, and thus minimizes spatial loss by using the same low-loss activation pattern through each layer. Plots are available in the appendix of the revised paper (**Figure 11**).

---

> ### Author Response · Authors · 2024-12-03
> **Summary of all addressed points**
>
> Thank you again for your valuable feedback that helped us to significantly improve our paper.
>
> Specifically, we believe we have addressed your main points in the following way
> - Effect of random permutations of unit locations has been demonstrated with an additional analysis showing the activation pattern propagates through the network (for details, see footnote 1 and fig. 12)
> - Multi-head attention might have benefits over single-head attention in modeling human neural activity. This point has been addressed at the end of the last paragraph of the section "related work".
> - We evaluated Topoformer-BERT as a baseline and show evaluations on all benchmarks. For details, see figs. 13 & 14 and the paragraph "Clustering in Topoformer-BERT" in both sections 5.1 and 5.2.
>
> In the light of these substantial improvements of the manuscript, we would be grateful to the reviewer if she/he could reassess her/his evaluation of our TopoLM paper.

---

> > ### Comment · Reviewer_3QeR · 2024-12-03
> >
> > I thank the authors for thoroughly addressing the questions posed in my review and adding further substantial experiments to the manuscript. I will raise my score to 8.

---

### Author Response · Authors · 2024-11-24
**General response #1**

We are very grateful to all 3 reviewers for their critical and very constructive evaluation of our work on TopoLM, a topographic language model showing hallmarks of language processing in the human brain. With this first response, we are addressing a number of important points raised by the reviewers, while other points will be addressed as soon as the associated additional analyses are completed.

Multiple reviewers suggested a comparison against another topographic baseline model. The version of our paper being evaluated here contains comparisons of TopoLM with its non-topographic equivalent (“non-topographic”), in which the spatial correlation loss term is switched off during training. We are currently working on evaluating the only other existing topographic language model (Topoformer BERT; BinHuraib 2024, https://openreview.net/forum?id=R6AA1NZhLd) on the main datasets we presented in the paper (Fedorenko 2024, fig. 2; Hauptman 2024, fig. 3; Moseley 2014, fig. 4) and will add the results in our response as soon as possible.

We address the other points raised by the reviewers in individual responses below.

---

> ### Author Response · Authors · 2024-11-26
> **General Response #2 — Topoformer Comparison**
>
> Thank you again for the helpful feedback! Reviewers U3qv and 3QeR both suggested a comparison between TopoLM and Topoformer (BinHuraib et al., 2024). We agree that this provides important context. We ran BinHuraib et al.’s Topoformer-BERT out-of-the-box on the three topographic neural benchmarks used in our paper (Fedorenko et al., 2010, Hauptman et al., 2024, and Moseley and Pulvermuller, 2014), results are below (and in the updated paper). We examine outputs at the attention layer and after multiplication by the key matrix, as in BinHuraib et al. 2024, as Topoformer induces local connectivity at both of these levels. We present all of this with the caveat that Topoformer-BERT is a baseline, not a control model: it is trained on a much smaller corpus (BookCorpus), is bidirectional, and has only one attention head.
>
> We find that for the Hauptman et al. (2024) and Moseley and Pulvermuller (2014) stimuli (noun-verb contrasts), Topoformer-BERT does not exhibit significant brain-like spatio-functional organization, both with attention and key matrix outputs, with few units coming out significant across contrasts (10.61% in Hauptman noun-verb, 0% in Moseley concrete-concrete and abstract-abstract). This is not to say that the model does not learn topography—indeed,  **before** applying a significance threshold, Topoformer-BERT exhibits a high degree of clustering (e.g. on Hauptman et al.: Moran’s I = 0.62 before fMRI readout sampling, I = 0.84 after). Rather, the topography of the model simply does not match the topography of the brain in terms of featural organization.
>
> Topoformer-BERT fares better with the Fedorenko et al. (2010) benchmark (sentences vs. nonword strings). We are able to successfully localize a spatially organized language network in the model, and most units come out significant (after the key matrix). This is likely in part because Topoformer-BERT induces topography only in the attention layers, and attention primarily tracks relationships between tokens: while the Fedorenko et al. stimuli consists of entire sentences, Hauptman et al. and Moseley and Pulvermuller’s stimuli are one or two word items. **More details (including plots) are in the revised version of the paper; see Figure 12.**
>
> However, it’s important to note that—while we think this comparison is a useful baseline for making sense of our results—the goal of our paper is to ask whether the TDANN spatial smoothness principle generalizes to language, not to build a “better” topographic language model. In other words, TopoLM is an in silico model of how spatio-functional organization emerges in the brain, and the fact that it predicts this organization well is evidence for TDANN spatial smoothness as a guiding principle for whole-cortex topography. Topoformer asks a similar question using a different guiding principle (local connectivity), but these models do not necessarily “compete” with one another: in particular, TopoLM’s spatial smoothness and Topoformer’s explicit local connectivity are both closely related to the principle of wiring length minimization.

---

> > ### Author Response · Authors · 2024-11-28
> > **General Response #3**
> >
> > In our final general response to the reviewers we want to take the opportunity to summarize our responses and the changes we made to the paper.
> >
> > In the latest version of our submission, we address the requested comparison between TopoLM and Topoformer Bert quantitatively. In addition to figure 13 (showkng Topoformer BERT maps for the Hauptman (2024) and Moseley & Pulvermüller (2014) analyses), we added figure 14 introducing a variant of Moran's I which in our eyes is more adequate to estimate autocorrelation in the maps produced by Topoformer BERT. More specifically, we had applied and are still applying Standard Moran's I on unthresholded maps throughout the paper which reflects clustering of the overall map well if few or no units show insignificant t-values (as is the case for TopoLM). In an attempt to accommodate the maps from Topoformer BERT containing mostly insignificant t-values (see fig. 13), for each layer's map we computed Moran's I for each "island" of significant t-values to then average values across islands to obtain a single autocorrelation estimate per map ("Islands Moran's I"). This analysis suggests no clustering or even dispersion in Topoformer BERT maps in the case of Hauptman 2024 while TopoLM shows quantitatively very similar results to when Standard Moran's I is used to estimate autocorrelation (fig. 14A). Due to a lack of significant units Islands Moran's I cannot be computed for Moseley & Pulvermüller 2014 (fig. 14B-E).
> >
> > Addressing weakness 5 rasied by r3 (U3qv), we performed the brain alignment analysis (brainscore language) using representational similarity analysis (fig. 6) instead of linear predictivity (fig. 5). While TopoLM outperforms its non-topographic counterpart on one benchmark (fig. 6, Pereira2018), it achieves lower values on two other benchmarks (Blank2014, Fedorenko2016). We believe that an overall brainscore language RSA result in favor of TopoLM's non-topographic counterpart needs to be interpreted with caution, because it is mainly driven by one benchmark (Fedorenko2016) with very noisy data, potentially inflating its effect on the overall score.
> >
> > While we wanted to include the discussion on the relation between our TDANN spatial correlation loss and factors ultimately leading to local correlations in the brain (see response to weakness 4-5 https://openreview.net/forum?id=aWXnKanInf&noteId=8BBoFXsjDf), we regret not having included this discussion in the paper due to space constraints. Unfortunately, we were not able to complete the the covariance analysis suggested by r3 (U3qv), weakness 2, in time.
> >
> > We thank the reviewers again for their very constructive feedback and are looking forward to a discussion of the paper over the next days. Except for the two points just mentioned, we believe we were able to address all points raised by the reviewers. In the light of the substantial improvements we were able to make with their help, we would be very grateful to the reviewers if they could reassess their evaluation of the TopoLM paper.

---

### Author Response · Authors · 2024-12-03
**Summary of reviewer feedback and request for area chair consideration**

Dear area chairs,

please find below a summary of the points raised by the reviewers and how we responded to them.

## Response to reviewer 3QeR

In our latest response to reviewer 3QeR (https://openreview.net/forum?id=aWXnKanInf&noteId=yz1b8Rc7DQ), we have summarized the  points we addressed:
- Evaluation of Topoformer BERT as a baseline topographic language model (same point requested by the other 2 reviewers)
- Effect of random permutations of unit locations on topographic organization
- The role of multi- vs. single-head attention

As a result, reviewer 3QeR raised their score from 6 to 8.

## Response to reviewer TBeb

In our latest responses to reviewer TBeb (https://openreview.net/forum?id=aWXnKanInf&noteId=v48FMPue7b), we have summarized the  points we addressed:
- Evaluation of Topoformer BERT as a baseline topographic language model (same point requested by the other 2 reviewers)
- Detailed architectural and training decisions
- Topography in TopoLM and the brain

Since we believe we have addressed the main points reviewer TBeb had raised, we have asked for a reassessment of their evaluation of our paper; we have not received a response to our request at the time of writing.

## Response to reviewer U3qv

Here, we are summarizing the  points we addressed in multiple responses to reviewer U3qv:
- **Evaluation of Topoformer BERT** as a baseline topographic language model (same point requested by the other 2 reviewers)
- **Objective function and biological implementation:** Relation between the spatial correlation loss and reduced wiring length
- **Discussion on whether cluster formation are surprising**
- **Clarification** with regard to the second condition under which we see the spatio-functional alignment between brain and model
- **Statistical analysis:** Uni- vs. multivariate
- **Implementation of spatial correlation loss**
- **Potential recursiveness of the core language system results**
- **Clarification** of claim on unified spatial objective in the conclusion
- **Relation between our TDANN spatial correlation loss and factors ultimately leading to local correlations in the brain**
- **Brain alignment** analysis using representational similarity instead of linear predictivity.

As a result, reviewer U3qv raised their score from 6 to 8.

We hope that the area chair considers this context in their evaluation of the TopoLM paper.

Thank you!

---

### Meta-Review · Area_Chair_KTqy · 2024-12-20

**Metareview:**

This paper modifies the standard autoregressive setting by adding a regularizing term that encourages nearby parts the model to behave similarly, with the motivation that this mimics known brain function. All reviewers agree that the idea is interesting and that the paper is ably executed.

**Additional Comments On Reviewer Discussion:**

See the thorough comments from reviewer U3qv, and the author responses.

---

### Decision · Program_Chairs · 2025-01-22

Accept (Oral)